# PACT-mediated PKR activation acts as a hyperosmotic stress intensity sensor weakening osmoadaptation and enhancing inflammation

Kenneth T Farabaugh[1], Dawid Krokowski[2,3], Bo-Jhih Guan[2], Zhaofeng Gao[2], Xing-Huang Gao[2], Jing Wu[2], Raul Jobava[4], Greeshma Ray[5], Tristan J de Jesus[6], Massimiliano G Bianchi[7], Evelyn Chukwurah[2], Ovidio Bussolati[7], Michael Kilberg[8], David A Buchner[2,4], Ganes C Sen[5], Calvin Cotton[9], Christine McDonald[5], Michelle Longworth[5], Parameswaran Ramakrishnan[6]*, Maria Hatzoglou[2]*

[1]Department of Pharmacology, Case Western Reserve University, Cleveland, United States; [2]Department of Genetics and Genome Sciences, Case Western Reserve University, Cleveland, United States; [3]Department of Molecular Biology, Maria Curie-Sklodowska University, Lublin, Poland; [4]Department of Biochemistry, Case Western Reserve University, Cleveland, United States; [5]Department of Inflammation and Immunity, Cleveland Clinic Foundation, Cleveland, United States; [6]Department of Pathology, Case Western Reserve University, Cleveland, United States; [7]Department of Medicine and Surgery, Universita degli Studi di Parma, Parma, Italy; [8]Department of Biochemistry and Molecular Biology, University of Florida, Gainesville, United States; [9]Department of Physiology and Biophysics, Case Western Reserve University, Cleveland, United States

*For correspondence:
pxr150@case.edu (PR);
mxh8@case.edu (MH)

Competing interests: The authors declare that no competing interests exist.

**Abstract** The inability of cells to adapt to increased environmental tonicity can lead to inflammatory gene expression and pathogenesis. The Rel family of transcription factors TonEBP and NF-κB p65 play critical roles in the switch from osmoadaptive homeostasis to inflammation, respectively. Here we identified PACT-mediated PKR kinase activation as a marker of the termination of adaptation and initiation of inflammation in *Mus musculus* embryonic fibroblasts. We found that high stress-induced PACT-PKR activation inhibits the interaction between NF-κB c-Rel and TonEBP essential for the increased expression of TonEBP-dependent osmoprotective genes. This resulted in enhanced formation of TonEBP/NF-κB p65 complexes and enhanced proinflammatory gene expression. These data demonstrate a novel role of c-Rel in the adaptive response to hyperosmotic stress, which is inhibited via a PACT/PKR-dependent dimer redistribution of the Rel family transcription factors. Our results suggest that inhibiting PACT-PKR signaling may prove a novel target for alleviating stress-induced inflammatory diseases.

## Introduction

The cellular response to increased extracellular osmolytes is a tightly regulated fundamental process that dictates the cell fate (*Burg et al., 2007*). Multiple signaling pathways are involved in orchestrating the hyperosmotic stress response, with the ultimate goal of maintaining a constant cell volume. Although the initial sensor of hyperosmolarity is unknown in mammalian cells, adaptation to hyperosmotic stress has been shown to be dependent on induction of genes by the transcription factor Tonicity Enhancer Binding Protein (TonEBP) (*Burg et al., 2007*). TonEBP translocates to the nucleus

**eLife digest** Cells are sensitive to changes in their environment. For example, maintaining normal salt levels in the blood, also called tonicity, is essential for the health of individual cells and the organism as a whole. Tonicity controls the movement of water in and out of the cell: high levels of salt inside the cell draw water in, while high levels of salt outside the cell draw water out. If salt levels in the environment surrounding the cells become too high, too much water will be drawn out, causing the cells to shrink.

Changes in tonicity can cause the cell to become stressed. Initially, cells adapt to this stress by switching on sets of genes that help restore fluid balance and allow the cell to regain its normal shape and size. If the increase in tonicity exceeds tolerable stress levels and harms the cell, this initiates an inflammatory response which ultimately leads to cell death. However, it remained unclear how cells switch from adapting to responding with inflammation.

Now, Farabaugh et al. have used an experimental system which mimics high salt to identify the mechanism that allows cells to switch between these two responses. The experiments showed that when salt levels are too high, cells switch on a stress sensing protein called PACT, which activates another protein called PKR. When PACT was deleted from mouse cells, this led to a decrease in the activity of inflammatory genes, and prevented the cells from self-destructing.

Other proteins that are involved in the adaptive and inflammatory response are the NF-κB family of proteins and TonEBP. Farabaugh et al. found that under low intensity stress, when salt levels outside the cell are slightly too high, a family member of NF-κB works with TonEBP to switch on adaptive genes. But, if salt levels continue to rise, PACT activates and turns on PKR. This blocks the interaction between NF-κB and TonEBP, allowing another family member of NF-κB to interact with TonEBP instead. This switches the adaptive response off and the inflammatory response on.

There are many diseases that involve changes in tonicity, including diabetes, cancer, inflammatory bowel disease, and dry eye syndrome. Understanding the proteins involved in the adaptive and inflammatory response could lead to the development of drugs that help to protect cells from stress-induced damage.

in response to an increase in osmolarity to regulate transcription of target genes associated with osmoadaptation (*Miyakawa et al., 1999*). These include chaperone proteins (heat shock protein 70 [HSP70, *Hspa1a*]; *Woo et al., 2002*), enzymes involved in biosynthesis of neutral osmolytes (aldose reductase [AR, *Akr1b1*]; *Ferraris et al., 1994*), and membrane proteins that transport organic osmolytes such as amino acids proline and alanine across the plasma membrane (sodium-coupled neutral amino acid transporter [SNAT2, *Slc38a2*]; *Trama et al., 2002*). The SNAT2-mediated arm of the adaptive response is also dependent on the membrane trafficking function of PP1/GADD34 (*Krokowski et al., 2017*).

When the degree of hyperosmotic stress becomes too high to overcome, the adaptive response fails. The failure to adapt will lead to an increase in inflammatory signaling, followed by apoptosis (*Brocker et al., 2012*). We have previously determined in MEFs that extracellular osmolarities of 500 and 600 mOsm are representative of low stress intensity/adaptive response and high stress intensity/non-adaptive response, respectively (*Farabaugh et al., 2017*). Inflammation is known to be regulated by many signaling pathways, including but not limited to NF-κB signaling (*Liu et al., 2017*). An increase in hyperosmolarity is associated with an increase in NF-κB activation (*Németh et al., 2002*) and inflammation (*Müller et al., 2019*), as well as production and release of multiple proinflammatory cytokines, including IL8 (10), TNFα (*Lang et al., 2002*), IL1β (*Ip and Medzhitov, 2015*), and IL6 (*Igarashi et al., 2014*). Apoptosis occurs following exposure to hyperosmolarity beyond which adaptation is possible, the limit of which can vary between cell types but is sharply defined within each (*Burg et al., 2007*). Apoptosis following hyperosmotic stress is characterized by depolarization of mitochondria leading to an increase in the Bax/Bcl-2 ratio (*Michea et al., 2002*), as well as sometimes by clustering and internalization of TNFα receptors (*Rosette and Karin, 1996*). This potentially contributes to a number of human diseases, such as dry eye syndrome (*Lemp et al., 2011*), diabetes (*Stookey et al., 2004*), and inflammatory bowel disease (*Schwartz et al., 2009*).

We have previously shown that NF-κB-dependent signaling is pro-apoptotic in high-intensity hyperosmotic stress, in part via induction of inducible nitric oxide synthetase (iNOS) (*Farabaugh et al., 2017*). The NF-κB transcription factor family consists of five subunits, three of which have transactivation activity (RelA/p65, Rel/c-Rel, and RelB) (www.bu.edu/nf-kb). These subunits are extensively post-translationally modified and can homodimerize or heterodimerize to form unique complexes that fine-tune different transcriptional programs (*Oeckinghaus and Ghosh, 2009*). NF-κB p65 has been previously shown to play a role in both hyperosmotic stress and inflammation (*Liu et al., 2017*; *Németh et al., 2002*). NF-κB p65 is phosphorylated and accumulates in the nucleus in hyperosmotic conditions, and induces the transcription of a number of proinflammatory target genes, such as *Nos2* (*Jia et al., 2013*), *Il6* (*Farabaugh et al., 2017*), and *Tnfa* (*Iwata et al., 1999*), as well as caspase 3/9 activation (*Eisner et al., 2006*). NF-κB p65 is necessary for initiation of the inflammatory response in response to LPS (*McDonald et al., 1997*; *Hobbs et al., 2018*), for the development of Th17 cells via induction of the transcription factor RORγ (*Ruan et al., 2011a*), and for pathology in many diseases with inflammatory components, including inflammatory bowel disease (*Han et al., 2018*; *Schreiber et al., 1998*) and dry eye syndrome (*Tan et al., 2018*; *He et al., 2011*). Although deletion of NF-κB p65 is embryonically lethal in mice, simultaneous deletion of *Tnfa* leads to survival, and demonstrates that NF-κB p65 plays a role in the prevention of TNF-induced toxicity during development (*Doi et al., 1999*). The role of NF-κB c-Rel has not been examined in hyperosmotic stress, though it has been shown to have both proinflammatory (*de Jesús and Ramakrishnan, 2020*; *Liu et al., 2017*) and developmental (*Gilmore and Gerondakis, 2011*) functions. c-Rel is required for transcription of many genes necessary for immune system function, including *Irf4*, which promotes proliferation of B cells (*Grumont and Gerondakis, 2000*), *Bcl2l1* and *Bcl2A1*, which promote survival of B cells (*Owyang et al., 2001*), and *Il21*, which promotes development of Th17 cells (*Chen et al., 2010*). c-Rel-deficiency leads to altered phenotypes in many disease models: c-Rel KO mice are resistant to EAE, an autoimmune model of multiple sclerosis driven by Th17 cells (*Chen et al., 2011*); c-Rel KO mice fail to develop severe colitis induced by treatment with dextran sodium sulfate (DSS)(*Luu et al., 2017*), perhaps also as a result of loss of Th17 cell development (*Chen et al., 2011*); and overexpression of c-Rel has been shown to protect human islets from cell death mediated by the cell death protein PDCD4 via an increase in microRNA-21 expression (*Ruan et al., 2011b*).

We have shown that protein kinase R (PKR) promotes the nuclear localization of Serine-536 phosphorylated NF-κB family member p65, which leads to an increase in inflammatory gene induction and cell death (*Farabaugh et al., 2017*). PKR is activated under hyperosmotic stress (*Farabaugh et al., 2017*), and it has also been shown previously to regulate NF-κB signaling in inflammatory conditions, via the direct phosphorylation of the inhibitor of κB (IκB; *Kumar et al., 1994*) or interaction with the inhibitor of κB kinase (IKK; *Ishii et al., 2001*). PKR activation and downstream NF-κB signaling were also evident after high intensity hyperosmotic treatment with dextran sodium sulfate (DSS, 10%) in cell culture (*Farabaugh et al., 2017*). In contrast, we observed adaptation to low intensity hyperosmotic treatment (5% DSS)(*Farabaugh et al., 2017*). PKR consists of two repeated double-stranded RNA (dsRNA)-binding motifs, DRBM I and DRBM II, and a C-terminal kinase domain (*Clemens and Elia, 1997*). PKR phosphorylates Ser-51 of the eukaryotic translation initiation factor 2 α subunit (eIF2α), contributing to inhibition of global mRNA translation (*Saelens et al., 2001*; *Bevilacqua et al., 2010*). Although PKR is classically activated by the binding of viral double-stranded RNA (dsRNA) to the dsRNA-binding domain, we have shown that the binding of dsRNA is unnecessary for PKR activation and downstream eIF2α phosphorylation in hyperosmotic conditions (*Farabaugh et al., 2017*).

Another potential mechanism of PKR activation is via protein–protein binding of PKR and the PKR-activating protein (PACT)(*Patel and Sen, 1998*). PACT is a dsRNA-binding protein that can bind PKR at several domains including the two dsRNA-binding domains and the N-terminal portion of the kinase domain (*Peters et al., 2001*). PACT interaction with the kinase domain of PKR, inhibits its closed conformation and promotes an open, active conformation (*Li et al., 2006*). In addition to constitutive phosphorylations at Ser-18 and Ser-246, PACT can become additionally phosphorylated at Ser-287 in stress conditions such as arsenite treatment and oxidative stress, which leads to increased association with PKR and PKR activation (*Peters et al., 2006*). The involvement of PACT in hyperosmotic stress conditions has not been previously investigated.

Here, we show that hyperosmotic stress induces PACT-mediated PKR activation, which disrupts the association of NF-κB c-Rel with other Rel homology domain (RHD)-containing binding partners NF-κB p65 and TonEBP. The loss of NF-κB c-Rel/NF-κB p65 and NF-κB c-Rel/TonEBP heterodimers in hyperosmotic conditions leads to a reduction in osmoadaptive and pro-survival gene transcription and enhances TonEBP/NF-κB p65-dependent proinflammatory signaling (*Figure 1A*). We also show that differential gene expression programs based on hyperosmotic stress intensity persist across multiple cell types and treatments with various osmolytes, the results of which equate to a universal switch from adaptation to increased inflammatory signaling.

## Results

### PACT activates PKR and promotes proinflammatory gene expression in high-intensity hyperosmotic stress

In order to determine whether PKR activation in high intensity hyperosmotic conditions occurs via interaction with activating protein PACT, we first investigated the effects of PACT knockdown on PKR activation. We suppressed PACT expression in mouse embryonic fibroblasts (MEFs) by lentiviral delivery of an shRNA directed against PACT. In these stable knockdown shPACT MEFs, PACT protein levels were significantly reduced compared to MEFs infected with the control lentiviral vector (shCon) (*Figure 1B*). Decrease in *Prkra* led to a substantial reduction in PKR activation at high-intensity hyperosmotic stress (600 mOsm) as measured by PKR phosphorylation at Serine-451 (*Figure 1B*). This reduction in PKR activation correlated with the reduction of phosphorylation of PKR substrate eIF2α at serine-51. This suggests that PACT is necessary for PKR activation and downstream signaling in high intensity hyperosmotic stress conditions.

It has previously been shown that PKR and PACT interact at three distinct protein domains with varying degrees of specificity (*Peters et al., 2001*). To verify that PACT interacts with PKR in response to hyperosmotic conditions, we performed co-immunoprecipitation experiments in PKR KO MEFs that had been reconstituted with FLAG-tagged human PKR (*Youssef et al., 2015*). In unstressed or low-intensity stress conditions (500 mOsm sucrose or 5% w/v DSS), we found no PACT-PKR interaction; however, in high-intensity stress conditions (600 mOsm sucrose or 10% w/v DSS), PACT-PKR existed in a complex, suggesting stress intensity-induced binding (*Figure 1C*).

Several phosphorylation sites of PACT have been proposed as being necessary for PACT-mediated activation of PKR. Phosphorylation of Ser-287 occurs in response to various cellular stresses, and depends upon prior phosphorylation of Ser-246 (*Peters et al., 2006*). To address the importance of these phosphorylation sites in high intensity hyperosmotic stress conditions, we reconstituted PACT KO MEFs with FLAG-tagged PACT constructs. When PACT KO MEFs were reconstituted with PACT bearing a mutated phosphosite at residue 287 (S287A), or at its prerequisite phosphosite residue 246 (S246A), interaction with PKR was diminished compared to the wild type (WT) PACT control (*Figure 1—figure supplement 1A*). Lacking these phosphorylation sites, and the interaction with PKR that depends on the modification of these residues, PKR was unable to be activated as measured by its autophosphorylation at Thr-451 (*Figure 1—figure supplement 1B*). Together, these data indicate that in response to high-intensity hyperosmotic stress, PACT phosphorylation at residues Ser-246 and Ser-287 promote its interaction with PKR to activate its kinase activity.

To gain functional insight into the interaction of PACT with PKR, we examined cell fate in MEFs depleted of PACT. We hypothesized that if PACT was necessary to activate PKR in high stress intensity, a loss of PACT protein would demonstrate a corresponding decrease in high stress-induced apoptotic activity. We found that shPACT cells showed significantly reduced caspase-3 activity, suggesting that PACT has a pro-apoptotic role in high-intensity hyperosmotic stress conditions (*Figure 1D*), similar to PKR (*Farabaugh et al., 2017*). Combined with *Figure 1C*, these data demonstrate that at low intensity stress conditions, there is neither any increase in caspase-3 activity nor interaction between PACT and PKR; PACT-PKR interaction and increased apoptosis only occur in high-intensity hyperosmotic stress. This result suggests that PACT-PKR interaction promoted PKR activation leading to apoptosis.

We have previously shown that expression of proinflammatory genes such as iNOS and IL6 are induced in high-intensity hyperosmotic conditions, and are in part dependent on signaling

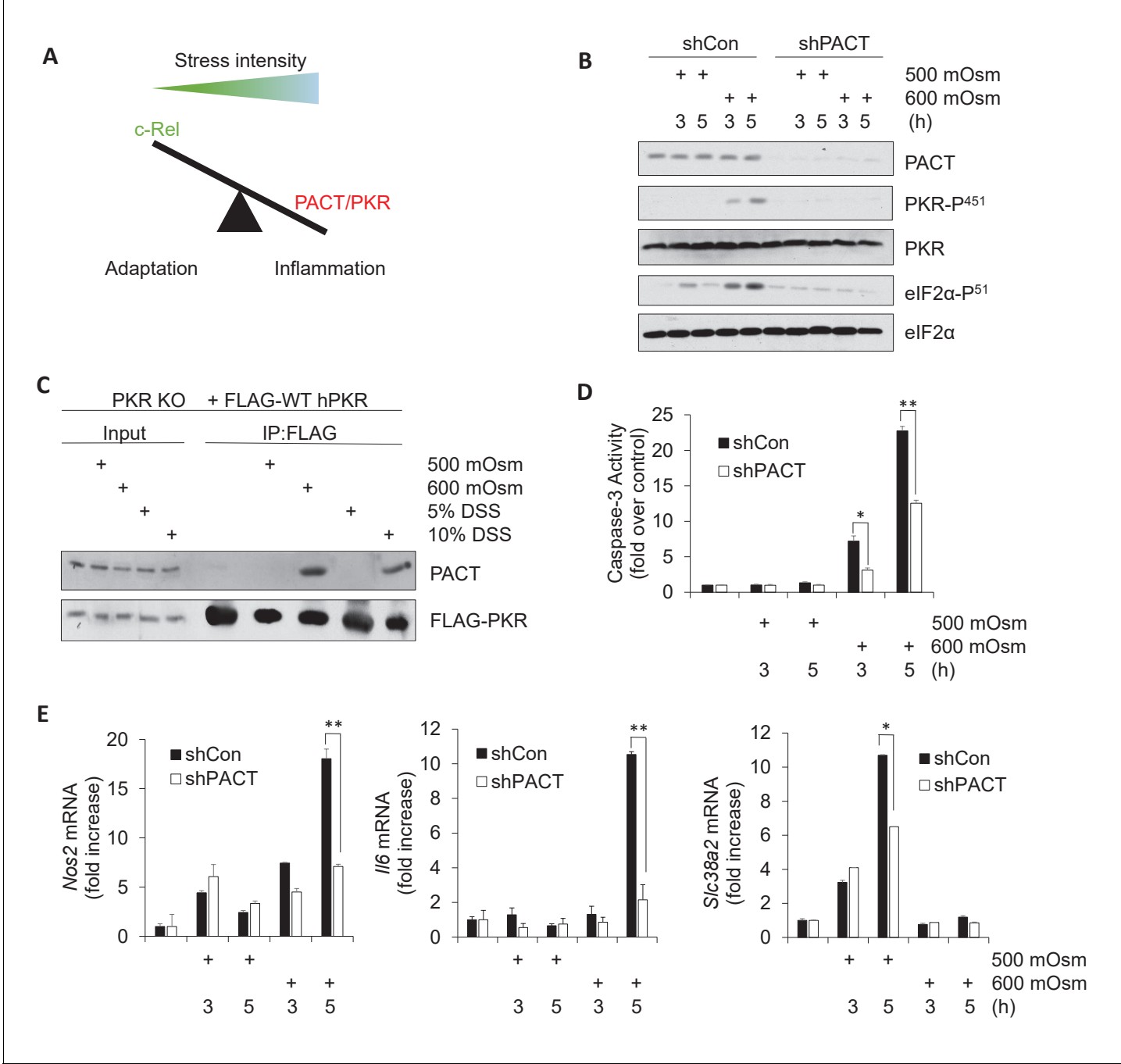

**Figure 1.** Activation of PACT-PKR axis promotes proinflammatory gene expression in response to high-intensity hyperosmotic stress. (**A**) As stress intensity increases, signaling shifts from c-Rel-mediated adaptation to PACT/PKR-mediated inflammation. (**B**) MEFs selected for shRNA-mediated knockdown of PACT and control vector were treated with 500 or 600 mOsm sucrose for the indicated durations. Lysates were analyzed via western blot for the indicated proteins. (**C**) PKR KO MEFs reconstituted with FLAG-tagged human PKR were treated with sucrose or DSS for 3 hr. Co-immunoprecipitation with FLAG antibody was analyzed by western blot. Inputs are 5% of immunoprecipitated sample. (**D**) Control and shPACT MEFs were treated with 500 or 600 mOsm sucrose. Lysates were assayed for caspase-3 enzymatic activity. (**E**) Control and shPACT MEFs were treated with 500 or 600 mOsm sucrose. RNA was isolated, and transcripts were analyzed via RT-qPCR for the indicated mRNAs.

The online version of this article includes the following source data and figure supplement(s) for figure 1:

**Source data 1.** Graph values for caspase-3 activity assays and RT-qPCR experiments in *Figure 1*.

**Figure supplement 1.** PACT Ser-246 and Ser-287 are necessary for PKR interaction and downstream inflammatory signaling.

**Figure supplement 2.** PACT influences the TonEBP-dependent adaptive transcription program.

downstream of PKR activation (*Farabaugh et al., 2017*). We hypothesized that if PACT played a role in PKR activation in hyperosmotic stress, the result of PACT knockdown would be a decrease in proinflammatory gene expression similar to the decrease observed in PKR-deficient cells (*Farabaugh et al., 2017*). We found that loss of PACT led to a decrease in mRNA induction of several proinflammatory genes such as *Nos2* and *Il6* under high intensity stress (*Figure 1E*). Further supporting the role of PACT-PKR signaling in high stress-induced inflammatory gene expression, we found that *Nos2* mRNA induction was also decreased in MEFs in which PACT phosphorylation at Ser-246 and Ser-287 was impaired (*Figure 1—figure supplement 1C*), and in which the kinase function of PKR was inactive by mutation of the catalytic lysine residue in the active site to arginine (*Figure 1—figure supplement 1D*). The ability of PKR to bind to dsRNA did not affect *Nos2* mRNA levels in hyperosmotic stress conditions, as demonstrated by impairment of RNA binding via double mutation of key residues involved in this interaction, Lys-64 and Lys-154 (*Figure 1—figure supplement 1D*). Together, these data demonstrate that functional PACT-PKR signaling is necessary for maximal expression of a subset of inflammatory genes. Finally, induction of osmoadaptive *Slc38a2* mRNA levels was unaffected after 3 hr of low-intensity stress treatment in cells depleted of PACT (*Figure 1E*); the decrease after 5 hr of low-intensity treatment may be a result of downstream PACT signaling on post-transcriptional mechanisms on *Slc38a2* mRNA accumulation. This decrease in osmoadaptive mRNA levels did not lead to caspase-3 activity or apoptosis under low intensity hyperosmotic stress (*Figure 1D*).

## NF-κB c-Rel promotes the adaptive response to hyperosmotic stress

Given the crucial role of the TonEBP transcription program for osmoadaptation (*Burg et al., 2007*), we next examined the effects of PACT on the nuclear localization of TonEBP in high and low stress intensity. Although it is well established that TonEBP homodimers induce expression of osmoadaptive genes (*Burg et al., 2007*), it has also been reported that TonEBP can form complexes with NF-κB p65 via the shared Rel homology domain (*Lee et al., 2016*). Since we have also shown that loss of PKR activation affects NF-κB nuclear accumulation in hyperosmotic conditions (*Farabaugh et al., 2017*), we speculated that the cellular context may influence TonEBP-mediated osmoadaptive gene expression. We therefore examined TonEBP expression and nuclear localization in shPACT MEFs treated with either high or low stress intensity. We found similar patterns in TonEBP nuclear localization upon PACT depletion either at low or at high stress intensity (*Figure 2A*). This indicates that TonEBP alone may not be sufficient to induce maximum levels of expression of osmoadaptive genes in response to hyperosmotic conditions irrespective of stress intensity. We therefore speculated that additional Rel homology domain family members may play a role in osmoadaptation dependent on PACT-mediated PKR signaling. Interestingly, we found nuclear accumulation of the NF-κB family member c-Rel at lower intensity in the absence of PACT (*Figure 2A*). In contrast to increased c-Rel, depletion of PACT decreases total NF-κB p65 and NF-κB p65-P(Ser-536) accumulation in the nucleus (*Figure 2A*), which may result in a decrease in NF-κB p65-mediated activation of proinflammatory genes. Although PACT-mediated PKR activation appears to decrease cell survival during high intensity hyperosmotic stress, a deficiency in PACT did not affect cell survival at low intensity hyperosmotic conditions, and correlated with an increase in nuclear localization of NF-κB c-Rel (*Figure 2A*).

While we have determined that NF-κB p65, particularly NF-κB p65 phosphorylated at Ser-536, is involved in increased apoptosis in high-intensity hyperosmotic conditions (*Farabaugh et al., 2017*), the role of NF-κB c-Rel was not known. To address this, we developed a stable knockdown MEF cell line deficient in NF-κB c-Rel. When treated with 500 mOsm medium, shc-Rel MEFs had higher levels of caspase-3 activity (*Figure 2B*), suggesting a pro-survival role for NF-κB c-Rel. If NF-κB c-Rel plays a role in the adaptive signaling pathway at this low intensity of stress, we hypothesized that its absence would inhibit osmoadaptive gene expression, and perhaps simultaneously promote proinflammatory signaling. Indeed, when cells are depleted of NF-κB c-Rel, NF-κB p65-P(Ser-536) accumulates in the nucleus at low intensity hyperosmotic stress (*Figure 2C*), a hallmark of the apoptotic signaling pathway previously identified (*Farabaugh et al., 2017*). This increase in NF-κB p65-P(Ser-536) correlates with an increase in iNOS mRNA levels (*Figure 2D*). These data suggest that NF-κB c-Rel is a novel factor in the transition from adaptation to proinflammatory gene expression programs with increasing stress intensity.

To further investigate the role of NF-κB c-Rel in low intensity stress, we examined in detail the adaptive program to hyperosmotic stress in the absence of NF-κB c-Rel. First we confirmed that the low

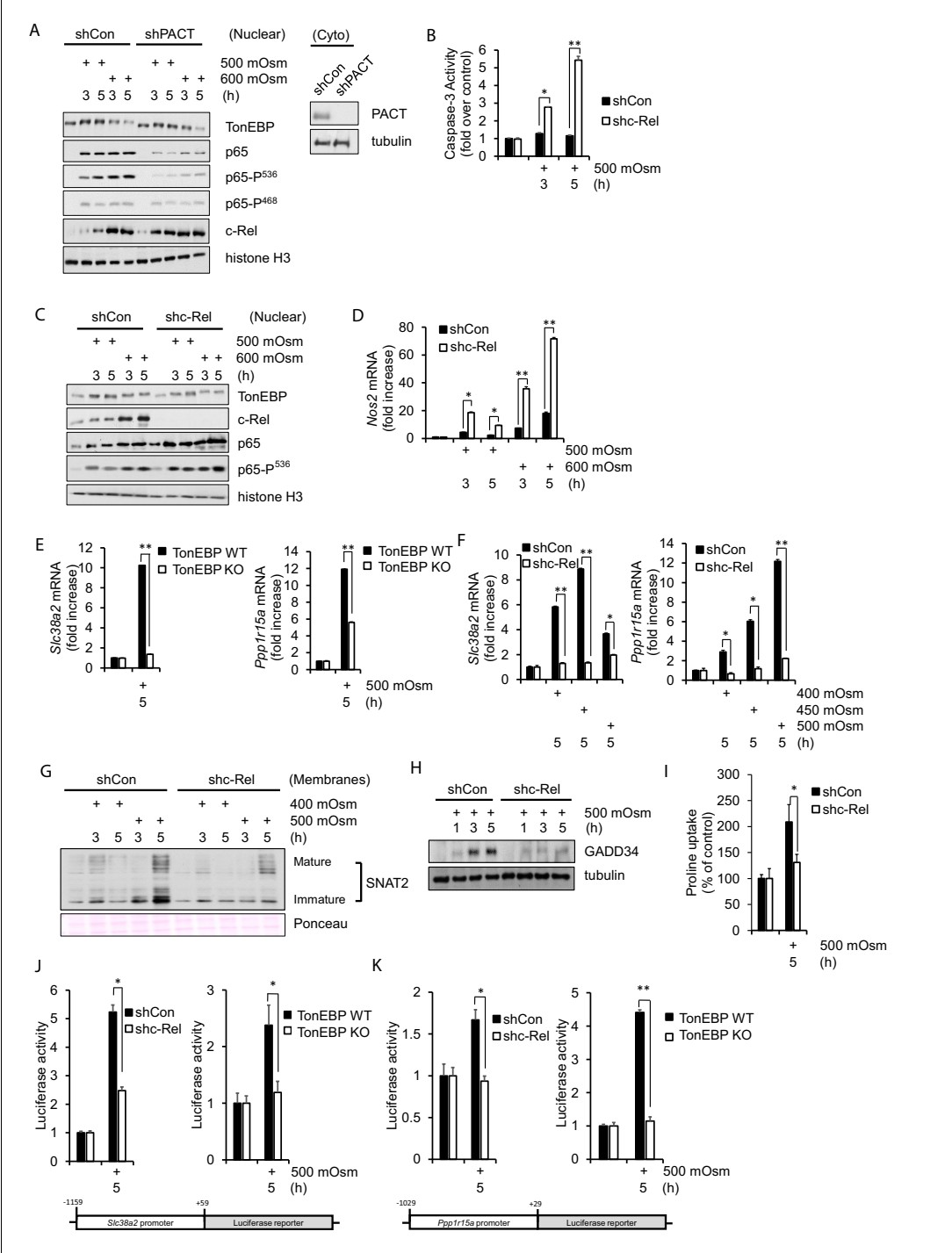

**Figure 2.** NF-κB c-Rel is a novel factor in the adaptive response to hyperosmotic stress. (**A**) Control and shPACT MEFs were treated with 500 or 600 mOsm. Nuclear fractions were isolated, and protein levels were analyzed via western blot. Insert indicates efficiency of PACT depletion from cytoplasmic extracts. (**B**) Control and shc-Rel MEFs were treated with 500 mOsm sucrose. Lysates were assayed for caspase-3 enzymatic activity. (**C**) Control and shc-Rel MEFs were treated with 500 or 600 mOsm sucrose. Nuclear fractions were isolated and protein levels were analyzed via western blot. (**D**) Control and shc-Rel MEFs were treated with 500 or 600 mOsm sucrose. RNA was isolated, and mRNA transcripts were analyzed via RT-qPCR. (**E**) TonEBP WT and TonEBP KO MEFs were treated with 500 mOsm sucrose. RNA was isolated, and mRNA transcripts were analyzed via RT-qPCR. (**F**) Control and shc-Rel MEFs were treated with varying intensities of sucrose. RNA was isolated, and mRNA transcripts were analyzed via RT-qPCR. (**G**) After indicated treatments, cytoplasmic membrane fractions were isolated and analyzed via western blot. (**H**) After indicated treatments, total lysates were isolated and analyzed via western blot. (**I**) Control and shc-Rel MEFs were treated with 500 mOsm sucrose. Intracellular levels of the amino acid proline were analyzed via amino acid uptake assay. (**J**) Control and shc-Rel MEFs and TonEBP WT and KO MEFs were transfected with the *Slc38a2-*

*Figure 2 continued on next page*

*Figure 2 continued*

promoter luciferase reporter construct, then treated with 500 mOsm sucrose. Luciferase activity was normalized to co-transfected Renilla luciferase activity. (**K**) Control and shc-Rel MEFs and TonEBP WT and KO MEFs were transfected with the *Ppp1r15a*-promoter luciferase reporter construct, then treated with 500 mOsm sucrose. Luciferase activity was normalized to co-transfected Renilla luciferase activity.

The online version of this article includes the following source data for figure 2:

**Source data 1.** Graph values for caspase-3 activity assays and RT-qPCR experiments in *Figure 2*.

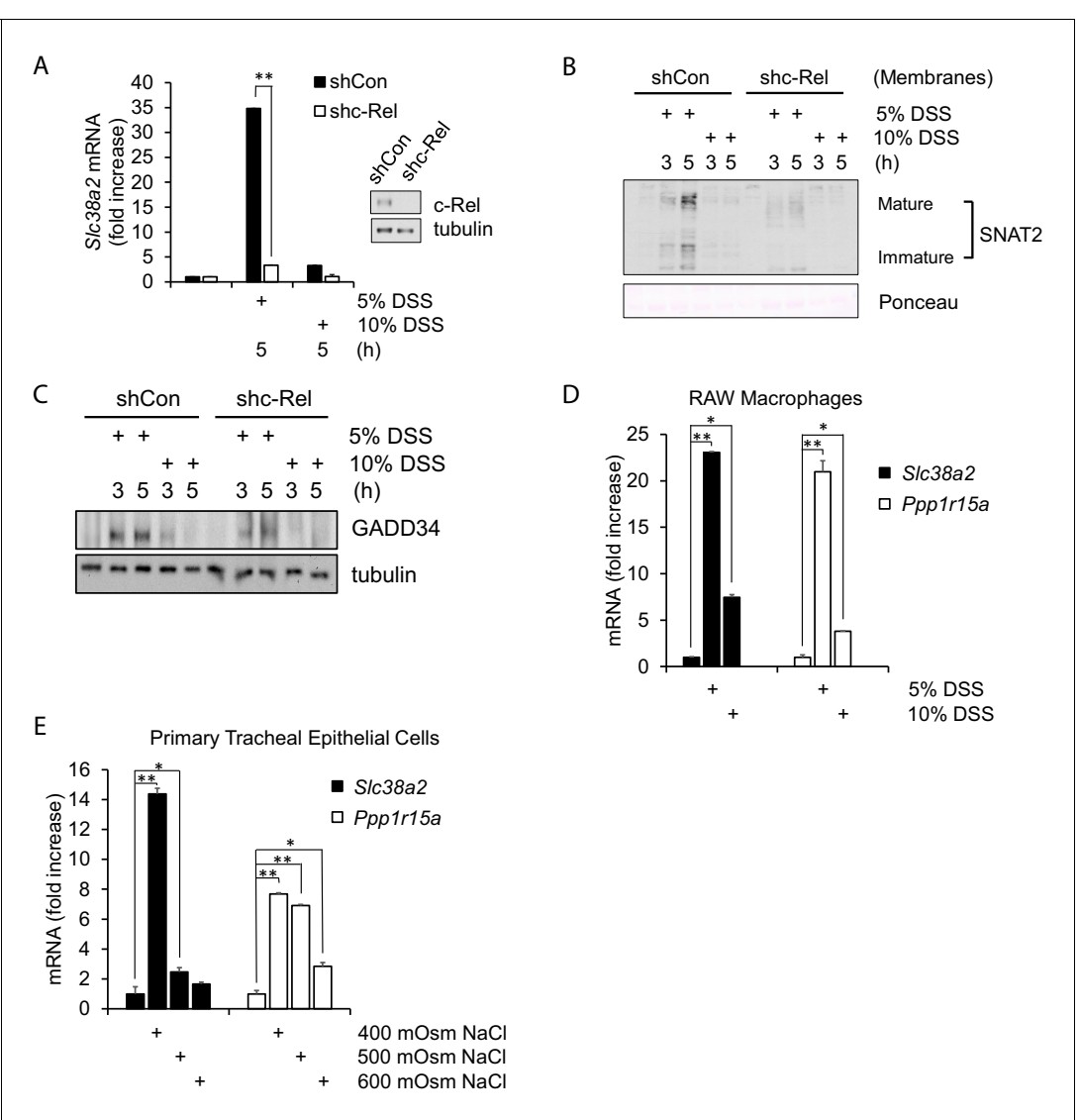

**Figure 3.** Hyperosmotic stress-induced adaptive gene expression occurs independent of cell type and osmolyte used. (**A**) Control and shc-Rel MEFs were treated with 5% or 10% DSS. RNA was isolated, and mRNA transcripts analyzed via RT-qPCR. (**B**) Control and shc-Rel MEFs were treated with 5% or 10% DSS. Cytoplasmic membrane fractions were isolated, and protein levels analyzed via western blot. (**C**) Control and shc-Rel MEFs were treated with 5% or 10% DSS. Total lysates were analyzed via western blot. (**D**) RAW macrophages were treated with 5% or 10% DSS for the indicated durations. RNA was isolated, and transcripts were analyzed via qPCR. (**E**) Primary tracheal epithelial cells cultured from human patients were treated with the indicated concentration of NaCl. RNA was isolated, and transcripts were analyzed via qPCR.

The online version of this article includes the following source data for figure 3:

**Source data 1.** Graph values for RT-qPCR experiments in *Figure 3*.

stress-mediated induction of osmoadaptive gene targets SNAT2 and GADD34 are dependent on TonEBP in our experimental system (*Figure 2E*). Next, we examined how loss of NF-κB c-Rel affects the low intensity stress-induced transcriptional induction of these adaptive mRNAs. We found that transcription of *Slc38a2* and *Ppp1r15a* is indeed diminished in NF-κB c-Rel-suppressed cells (*Figure 2F*). These data identify SNAT2 and GADD34 as gene targets not only of the TonEBP-mediated transcriptional response to hyperosmotic stress, but also of the NF-κB c-Rel-mediated adaptive response. To our knowledge, the dependence of osmoadaptation on NF-κB c-Rel is a new finding in the literature. The reduction in *Slc38a2* mRNA in shc-Rel MEFs led to reduced protein levels in both immature, unprocessed SNAT2 and the mature glycosylated form, as determined by western blot analysis of cytoplasmic membrane fractions (*Figure 2G*). GADD34 protein levels were also reduced in shc-Rel MEFs (*Figure 2H*). As expected, decreased protein levels of both SNAT2 and GADD34 resulted in decreased SNAT2 functionality, as measured via uptake of the amino acid proline (*Figure 2I*).

To further confirm the roles of both NF-κB c-Rel and TonEBP in mediating *Slc38a2* and *Ppp1r15a*-transcription, we performed luciferase reporter gene assays using constructs bearing the endogenous SNAT2 promoter from base pairs −1159 to +59 respective to the transcriptional start site (*Figure 2J*), or the endogenous GADD34 promoter from base pairs −1000 to +1 (*Figure 2K*). We found that the luciferase activity under control of both *Slc38a2* and *Ppp1r15a* promoters were much less responsive under hyperosmotic conditions in shc-Rel MEFs and TonEBP$^{-/-}$ MEFs compared to controls (*Figure 2J–K*). These results suggest that NF-κB c-Rel contributes to adaptation to hyperosmotic conditions at least in part via the SNAT2-dependent arm of the adaptive program.

## Loss of adaptation at high-intensity hyperosmotic stress conditions occurs independent of cell type and osmolyte

We demonstrated above that DSS treatment of cells in culture results in similar patterns of PACT-PKR interaction as hyperosmotic treatment with sucrose (*Figure 1C*), and that the PKR-mediated signaling promotes the inflammatory phenotype in DSS-treated mice (*Farabaugh et al., 2017*). In order to determine whether NF-κB c-Rel maintains the protective role in an inflammatory disease model, we turned to a cell-based model of DSS-induced hyperosmolarity. DSS has previously been shown to increase hyperosmolarity (*Farabaugh et al., 2017*), potentially in part via dissociation of the sodium moiety (*Laroui et al., 2012*). We hypothesized that if NF-κB c-Rel was protective in mild hyperosmotic stress conditions, that a loss of NF-κB c-Rel would result in increased expression of proinflammatory genes upon treatment of MEF cells with DSS and in mice at early time points of DSS-induced colitis. MEF cells were treated with media containing 5% or 10% DSS, as representatives of low and high stress intensity. We found that while wild type MEFs showed *Slc38a2* induction after a low intensity of DSS treatment, similar to the low-intensity sucrose treatment, while NF-κB c-Rel-deficient MEFs lost this induction of the adaptive *Slc38a2* mRNA (*Figure 3A*). This loss of SNAT2 mRNA induction in shc-Rel MEFs led to a decrease of both the unprocessed immature and mature forms of SNAT2 protein (*Figure 3B*). Also similar to sucrose-treated MEFs, DSS-treated shc-Rel MEFs show decreased GADD34 protein levels (*Figure 3C*).

To confirm that this response is a function of increased hyperosmolarity in general and not dependent on a specific cell type (MEFs) or osmolyte (sucrose), we also used DSS and NaCl treatment in RAW macrophages and human-derived tracheal epithelial cells. RAW macrophages treated with DSS display induction of osmoadaptive genes SNAT2 and GADD34 at low intensity treatment (*Figure 3D*). Primary tracheal epithelial cells also displayed this pattern of gene expression in response to low-intensity treatment with NaCl (*Figure 3E*), indicating that this gene expression program appears to be universal.

## Transcriptional differences in low-intensity and high-intensity hyperosmotic stress conditions

To identify the entire set of genes displaying differential expression based on hyperosmotic stress intensity, we performed whole-cell RNA sequencing in response to low or high intensity hyperosmotic treatment. Wild type MEFs were subjected to low (500 mOsm) or high (600 mOsm) intensity hyperosmotic stress, total RNA was isolated, and libraries were prepared and sequenced. Similar numbers of reads were obtained for each sample (*Figure 4—figure supplement 1A*), indicating consistency in sample preparation. 10,443 unique mRNAs were sequenced, and their expression profiles

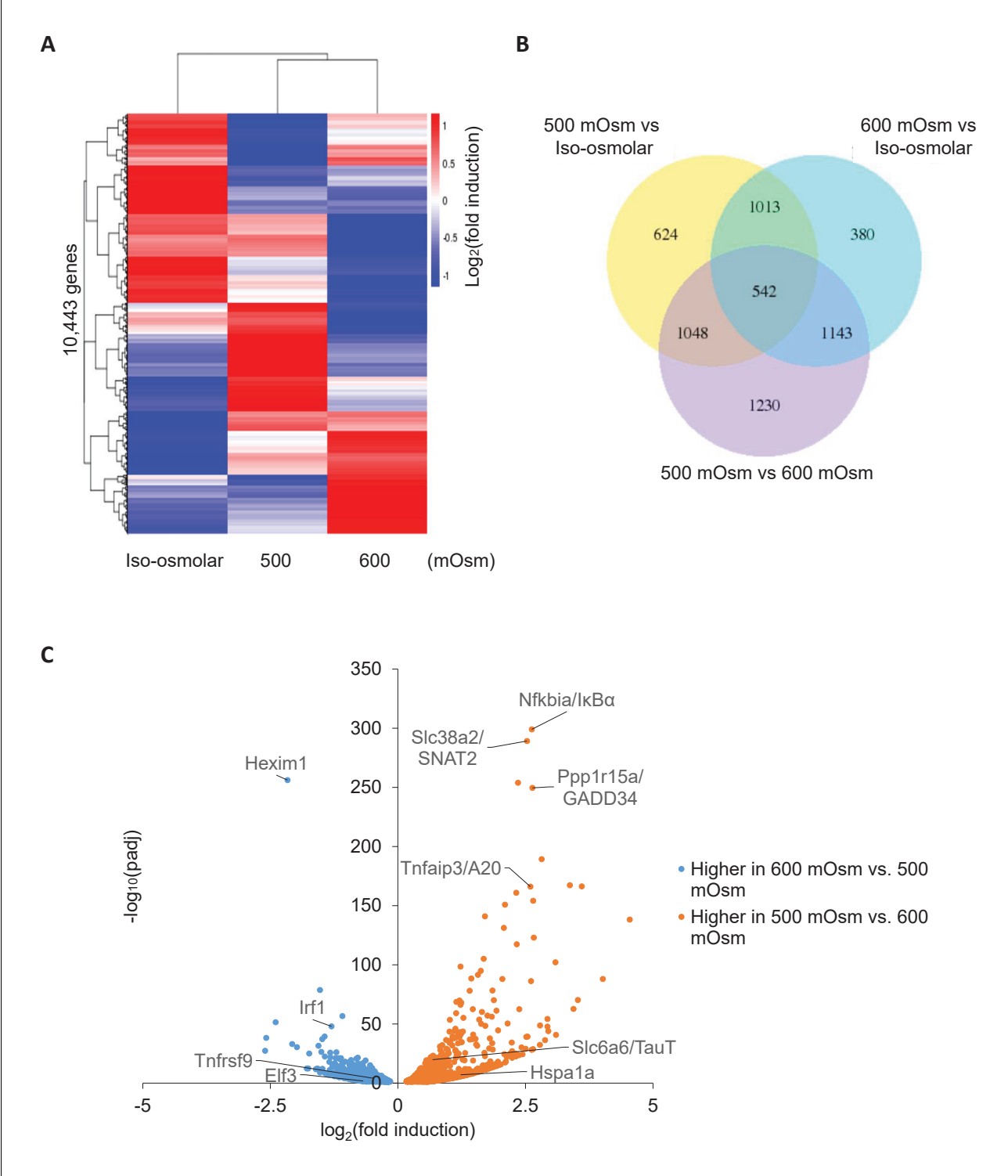

**Figure 4.** Differential gene expression in MEFs treated with low- or high-intensity hyperosmotic stress. (**A**) WT MEFs were treated with 500 or 600 mOsm sucrose for 3 hr. RNA was isolated, libraries prepared, and total RNAs sequenced on an Illumina platform. Heatmap shows differences in expression for 10,443 unique genes between each treatment. (**B**) Venn diagram of differentially expressed genes shared between control, low intensity, and high intensity-treated cells. (**C**) Volcano plot analysis of upregulated and downregulated genes between low intensity and high intensity-treated cells.

The online version of this article includes the following figure supplement(s) for figure 4:

**Figure supplement 1.** Hyperosmotic stress intensity alters expression of genes involved in ribosomal machinery and cancer signaling pathways.

were visualized in a heatmap (*Figure 4A*). We chose to focus on the genes differentially expressed between low-intensity and high-intensity treatment, so as to distinguish genes involved in the switch from the adaptive to the inflammatory phenotype. While 3963 total genes were differentially expressed ($p<0.05$) between low-intensity and high-intensity treatment, 2733 of these genes were also differentially expressed between iso-osmolar and hyperosmolar samples of both low and high intensity (*Figure 4B*), indicating a subset of genes differentially expressed in stress conditions independent of stress intensity (*Figure 4B*). These differentially expressed genes were further classified as being induced or inhibited from low-intensity to high-intensity stress, and KEGG pathway analysis was performed on these gene subsets. The most significant common pathways among genes induced at high-intensity but not at low-intensity hyperosmotic stress were associated with changes in the ribosome (n = 75, $p=8.40\times10^{-51}$) and oxidative phosphorylation (n = 31, $p=2.20\times10^{-9}$)(*Figure 4—figure supplement 1B*). The most significant increase ($p=6.68\times10^{-257}$) of a gene after high intensity treatment that was not increased at low intensity treatment occurred in Hexim1 (*Figure 4C*), a transcriptional elongation factor that has been shown to suppress tumor growth in response to nucleotide stresses (*Tan et al., 2016*). The most significant common pathways among genes induced at low-intensity but not at high-intensity were associated with various signaling pathways, including cancer (n = 76, $p=6.50\times10^{-11}$), and in focal adhesion (n = 39, $p=9.20\times10^{-6}$) and regulation of the cytoskeleton (n = 44, $p=1.6\times10^{-7}$) (*Figure 4—figure supplement 1B*). NF-κB signaling ($p=0.01$) and inflammatory bowel disease ($p=0.0047$) pathways were both significant in this subset of differentially expressed genes. However, the most significantly increased genes at low intensity versus high intensity have more direct relevance to the aforementioned novel osmoadaptive response, including GADD34 (*Ppp1r15a*) ($p=2.77\times10^{-250}$) and SNAT2 (*Slc38a2*) ($p=6.45\times10^{-290}$), as well as NF-κB pathway negative feedback inhibitors IκBα (*Nfkbia*) ($p=7.34\times10^{-300}$) and A20 (*Tnfaip3*) ($p=6.97\times10^{-167}$) (*Figure 4C*).

## NF-κB c-Rel/TonEBP target genes are involved in the adaptive response to hyperosmotic stress

As TonEBP is involved in the adaptive program, and has been shown to be involved in proinflammatory gene induction in other stress conditions (*Johnson et al., 2017*), we chose to examine in further detail the stress intensity-dependent expression pattern of select genes previously confirmed to be targets of TonEBP (*Izumi et al., 2015*; *Johnson et al., 2017*). We hypothesized that TonEBP target genes would be upregulated after low intensity hyperosmotic stress, but that this induction may be lost at high intensity stress conditions, similar to SNAT2 (*Figure 1E*). Even among the selected *bona fide* TonEBP targets, there was variability in expression patterns, in that many genes were upregulated only at low stress, similar to SNAT2, whereas others were upregulated only after high-intensity stress treatment (*Figure 5A*). We speculated that this shift in target gene expression may be related to the dimerization partner of TonEBP, as it has been demonstrated that TonEBP/NF-κB p65 complexes induce expression of several proinflammatory genes but not of osmoadaptive genes (*Johnson et al., 2017*). As we had identified *Slc38a2* and *Ppp1r15a* to be potential targets of both NF-κB c-Rel and TonEBP (*Figure 2D–E*), we proposed to identify other subsets of genes that were induced via specific complex makeup using NF-κB c-Rel-, NF-κB p65-, and TonEBP-deficient MEFs. Some TonEBP targets, including *Hspa1a* and *Mmp9*, displayed dependence on both TonEBP and NF-κB c-Rel (*Figure 5B*), hinting at the existence of a subset of genes that are targets of a TonEBP/NF-κB c-Rel complex. Other osmoadaptive TonEBP targets, such as aldose reductase (*Akr1b1*) and the taurine transporter (*Slc6a6*) were not influenced by NF-κB c-Rel (*Figure 5C*), indicating the regulation of a subset of these adaptive genes is NF-κB c-Rel-independent and likely are targets of TonEBP homodimers (*Stroud et al., 2002*). A third subset of genes displayed dependence on TonEBP and NF-κB p65 at high-intensity stress, including *Elf3*, *Irf1*, and *Tnfrsf9* (*Figure 5D*). We have previously shown that *Il6* mRNA levels are dependent on NF-κB p65 in hyperosmotic stress conditions (*Farabaugh et al., 2017*), and here also show that it is dependent on TonEBP (*Figure 5D*). We concluded that some genes (*Slc38a2*, *Mmp9*, *Hspa1a*, *Pppar15a*) were positive targets of NF-κB c-Rel/TonEBP, and others (*Il6*, *Elf3*, *Irf1*, *Tnfrsf9*) were positive targets of NF-κB p65/TonEBP (*Figure 5E*). These data suggest that there is competition between NF-κB p65 and NF-κB c-Rel for TonEBP interaction. We hypothesized that if loss of PACT significantly decreased nuclear accumulation of NF-κB p65, then NF-κB p65/TonEBP target gene transcription would be impaired at high-intensity hyperosmotic stress. These NF-κB p65/TonEBP targets that showed reduced

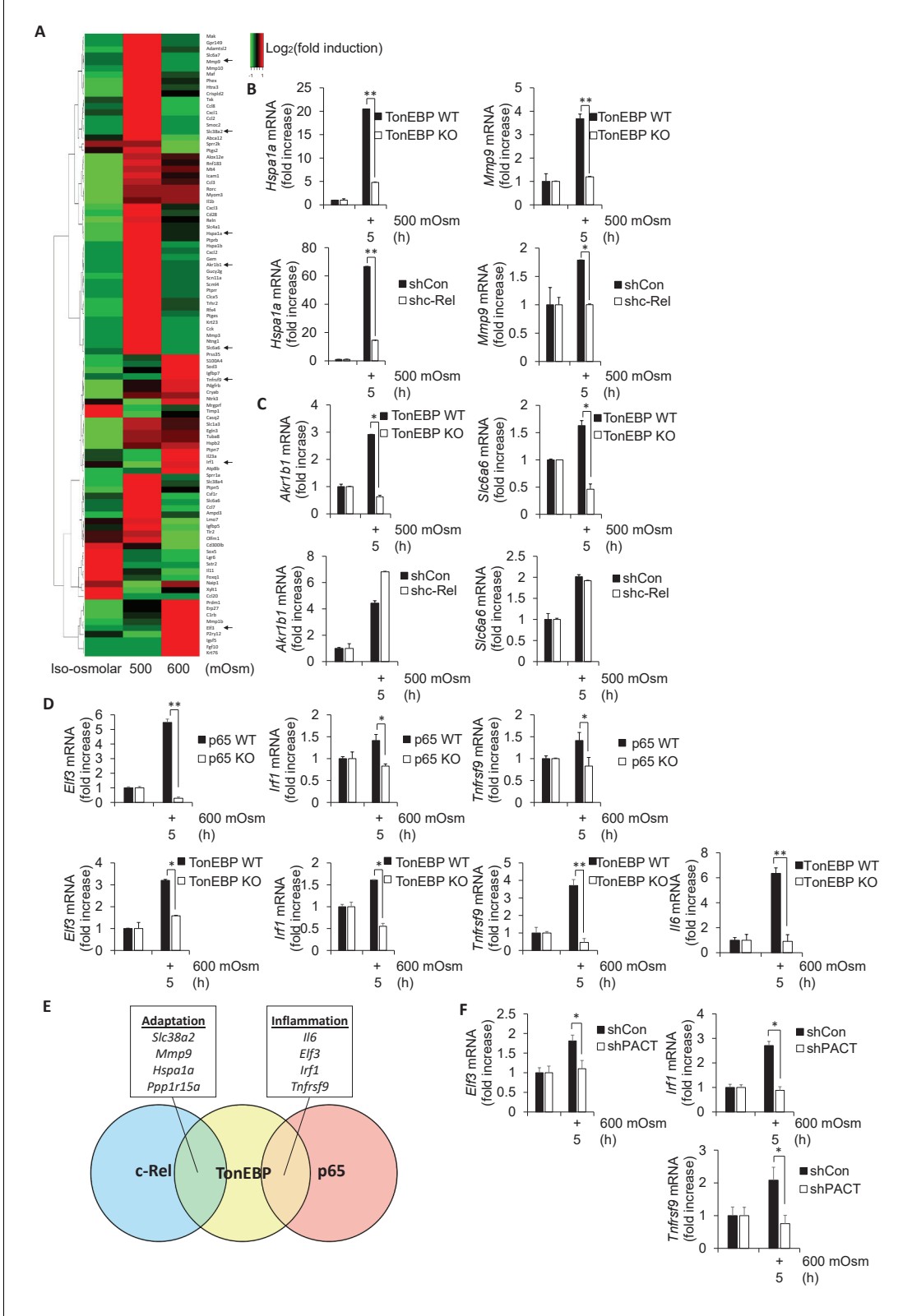

**Figure 5.** Subsets of genes induced by hyperosmotic stress are transcriptionally controlled by NF-κB c-Rel and TonEBP. (**A**) TonEBP targets identified by literature demonstrate a general increase at mild intensity and decrease at high intensity hyperosmotic stress. (**B–D**) MEFs deficient in NF-κB c-Rel, NF-κB p65, and TonEBP were treated with the indicated stress intensity. RNA was isolated, and mRNA transcript levels analyzed via RT-qPCR. (**E**)

*Figure 5 continued on next page*

*Figure 5 continued*

Genes were categorized based on targeting of Rel-homology domain transcription factor complexes. (F) MEFs deficient in PACT were treated with the indicated stress intensity. RNA was isolated, and mRNA transcript levels analyzed via RT-qPCR.

The online version of this article includes the following source data and figure supplement(s) for figure 5:

**Source data 1.** Graph values for RT-qPCR experiments in *Figure 5*.

**Figure supplement 1.** Additional proinflammatory gene expression programs are also dependent on PACT.

induction upon loss of PACT include *Il6* (*Figure 1E*), *Elf3*, *Irf1*, and *Tnfrsf9* (*Figure 5F*). Several additional proinflammatory genes, including *Ccl2*, *Cxcl1*, and *Icam-1*, were also discovered to have a dependence on PACT-mediated signaling, as evidenced by loss of their induction in PACT-deficient cells (*Figure 5—figure supplement 1A*).

## The PACT/PKR signaling axis inhibits the interaction of NF-κB c-Rel with either NF-κB p65 or TonEBP and increases interaction of NF-κB p65 with TonEBP

It has been previously shown that TonEBP/NF-κB p65 interaction promotes expression of proinflammatory targets of NF-κB p65, but not expression of osmoadaptive targets of TonEBP in response to TNF or LPS treatment (*Johnson et al., 2017*). We hypothesized that a similar interaction would occur under high-intensity hyperosmotic conditions (*Figure 6A*). To address this, we transfected MEFs with FLAG-tagged NF-κB p65, and co-immunoprecipitated protein complexes using a FLAG antibody. We observed that an increase in stress intensity increases the levels of NF-κB p65/TonEBP complexes (*Figure 6B*, right). Interestingly, there was an inverse correlation to complexes of NF-κB p65/NF-κB c-Rel, which were observed at iso-osmolar and low intensity-hyperosmotic conditions and lost upon high intensity-hyperosmotic treatment. We also found that activation of the PACT/PKR signaling axis promotes the loss of interaction of TonEBP with NF-κB c-Rel and the gain of interaction of TonEBP with NF-κB p65, as shPACT MEFs retain some NF-κB p65/NF-κB c-Rel complexes and NF-κB p65/TonEBP complex formation is diminished (*Figure 6B*). We observed that TonEBP/NF-κB p65 interaction still occurs in shPACT MEFs at high stress intensity; this suggests that PACT signaling determines the amplitude of the response, not merely whether the response occurs. This finding was corroborated by the inverse experiment, in which MEFs transfected with FLAG-c-Rel were also found to accumulate complexes of NF-κB c-Rel/NF-κB p65 and NF-κB c-Rel/TonEBP at iso-osmolar and low-intensity hyperosmotic conditions, and lose these complexes at high-intensity stress (*Figure 6C*).

Loss of PACT protein shifted the Rel-homology domain protein complexes from those present in the wild type at high-intensity stress towards resembling the composition of complexes present at a low intensity of stress. This correlates with a reduction in proinflammatory gene expression and apoptosis, which may suggest a protective function of TonEBP/NF-κB c-Rel and NF-κB p65/NF-κB c-Rel complexes. To confirm that the observed shift in Rel-homology domain complex composition is dependent on PACT via its function in PKR activation, we first used a pharmacological inhibitor of the kinase activity of PKR (PKRi), which we have described previously (*Farabaugh et al., 2017*). While FLAG-tagged NF-κB c-Rel did not pull down TonEBP or NF-κB p65 in high-intensity hyperosmotic conditions in untreated cells, PKRi treatment promoted the maintenance of these complexes even at high intensity hyperosmotic stress (*Figure 6D*). In addition, reconstitution of PACT KO MEFs with PACT bearing phosphosite mutations S246A or S287A demonstrated that loss of PACT functionality in PKR activation also shifted the makeup of Rel-homology domain complexes to resemble the composition in a low stress intensity (*Figure 6E–F*). Together, these data indicate that activation of the PACT/PKR axis promotes dissociation of NF-κB p65 and TonEBP from NF-κB c-Rel in a manner dependent on stress intensity.

## NF-κB c-Rel/TonEBP complexes accumulate in the nucleus in absence of PACT-PKR signaling

In response to cellular stress such as TNF or LPS treatment, NF-κB p65 has been reported to translocate to the nucleus (*Moreno et al., 2010*). We found that levels of nuclear NF-κB p65 and NF-κB c-Rel both increased upon hyperosmotic treatment, particularly at high intensity treatment (*Figure 7A*, left). Upon loss of NF-κB c-Rel, the percentage of nuclear NF-κB p65 over total cellular

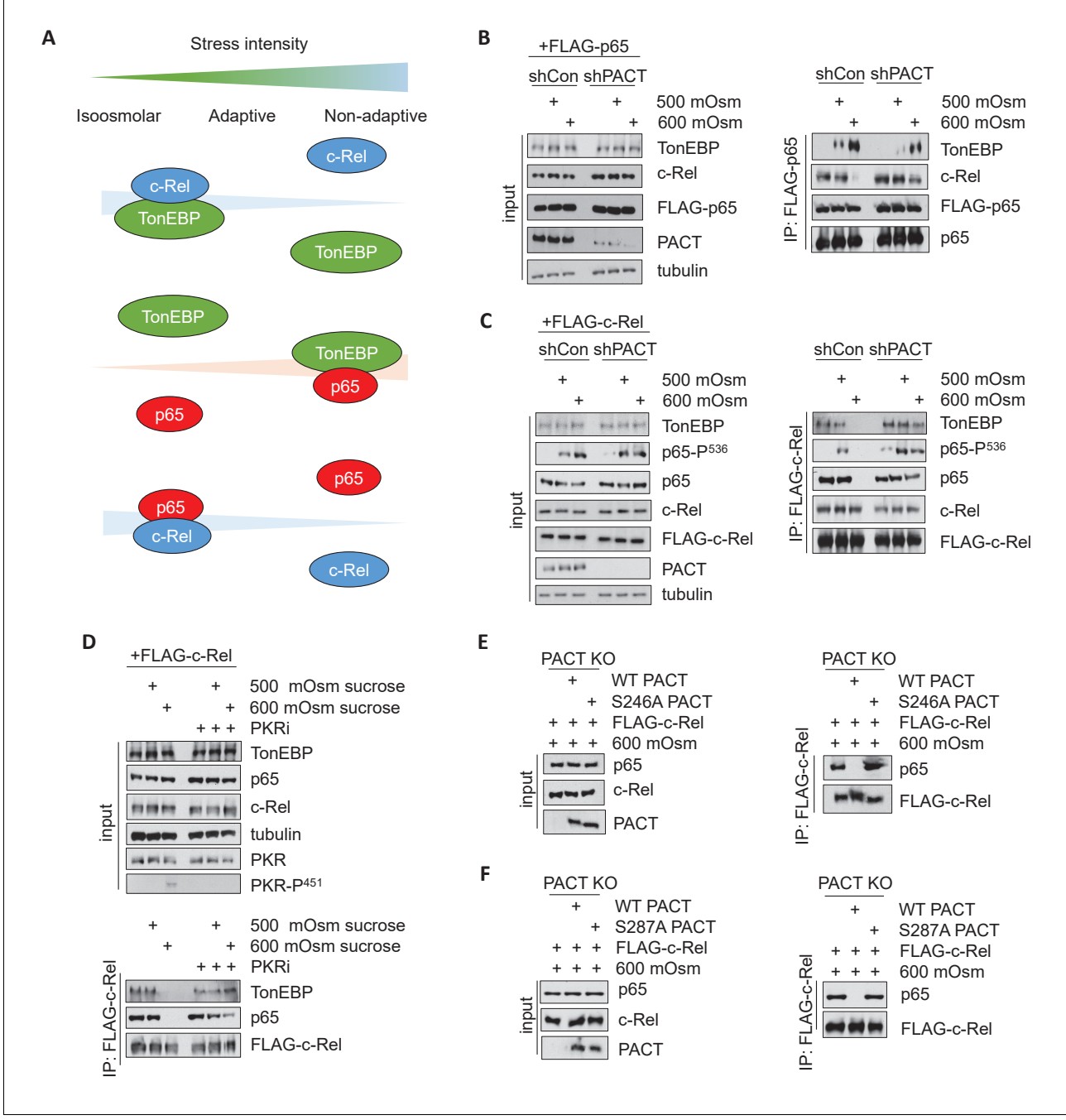

**Figure 6.** Activation of the PACT/PKR signaling axis inhibits the interaction of NF-κB c-Rel with NF-κB p65 or TonEBP, and increases the interaction of NF-κB p65 with TonEBP. (**A**) As stress intensity increases, NF-κB c-Rel/TonEBP and NF-κB c-Rel/NF-κB p65 species dissociate, while NF-κB p65/TonEBP species accumulate. (**B**) Control and shPACT MEFs were transfected with a FLAG-NF-κB c-Rel construct, then treated with 500 or 600 mOsm sucrose for 3 hr. Total cell extracts and FLAG-co-immunoprecipitated proteins were analyzed via western blot. (**C**) Control and shPACT MEFs were transfected with a FLAG-NF-κB p65 construct, then treated with 500 or 600 mOsm sucrose for 3 hr. Total cell extracts and FLAG-co-immunoprecipitated proteins were analyzed via western blot. (**D**) WT MEFs were transfected with a FLAG-NF-κB c-Rel construct, then treated with the indicated stress for 3 hr in the presence or absence of a small molecule inhibitor of PKR. Total cell extracts and FLAG-co-immunoprecipitated proteins were analyzed via western blot. (**E**) PACT KO MEFs were reconstituted with WT- or S246A-PACT, and transfected with a FLAG-NF-κB c-Rel construct. After treatment with 600 mOsm sucrose for 3 hr, complexes were co-immunoprecipitated with the FLAG antibody and analyzed via western blot, in parallel with total cell extracts. (**F**) PACT KO MEFs were reconstituted with WT- or S287A-PACT, and transfected with a FLAG-NF-κB c-Rel construct. After treatment with 600 mOsm sucrose for 3 hr, complexes were co-immunoprecipitated with the FLAG antibody and analyzed via western blot in parallel with total extracts.

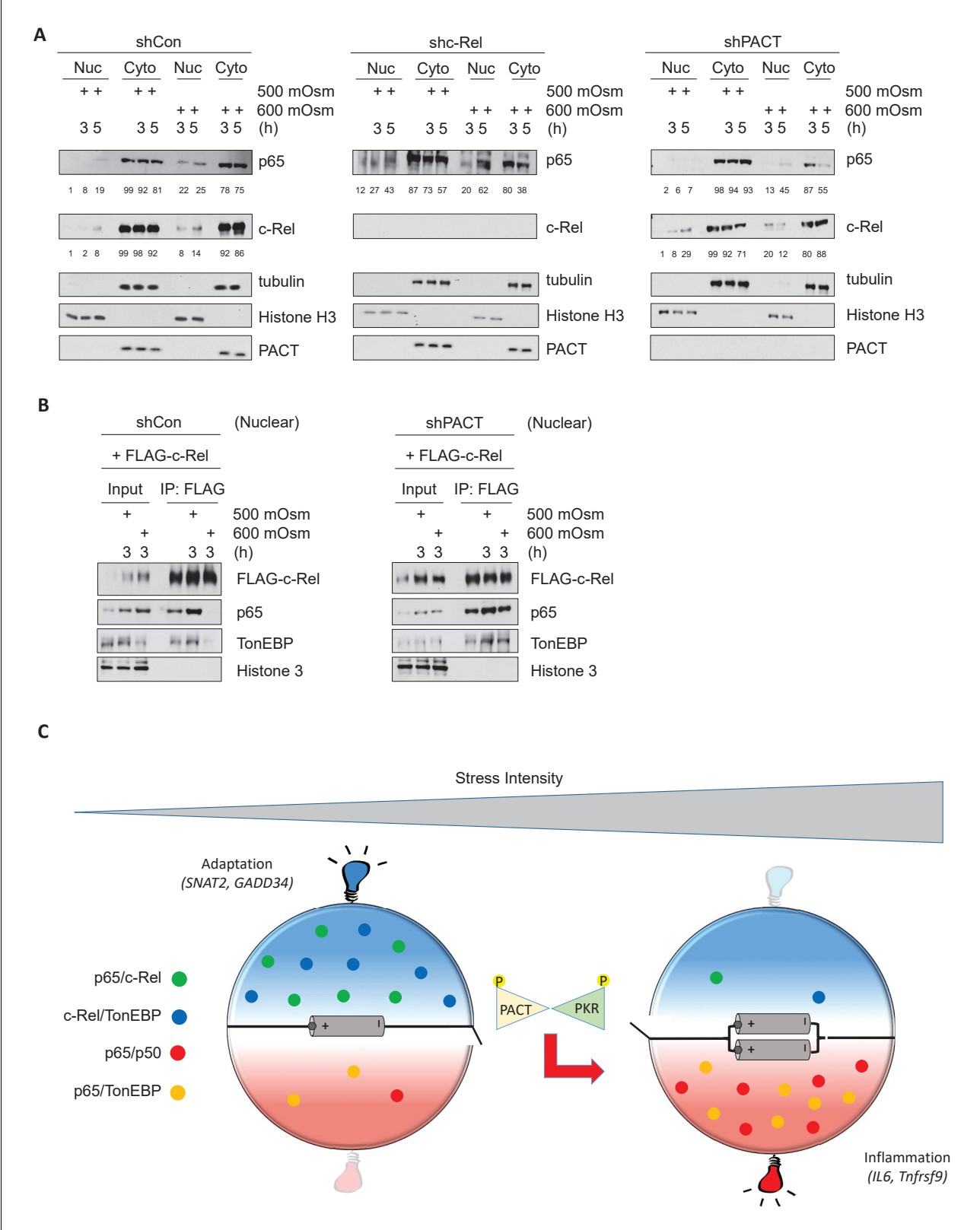

**Figure 7.** Hyperosmotic stress causes nuclear translocation of a small subset of NF-κB proteins. (**A**) Control, shc-Rel, and shPACT MEFs were treated with the indicated intensities and durations of hyperosmotic stress, then cells were fractionated into cytoplasmic and nuclear lysates. Equal volumes of protein were analyzed via western blot. Protein localization was quantified as percentages in the corresponding nuclear or cytoplasmic fractions by

*Figure 7 continued on next page*

*Figure 7 continued*

densitometric analysis. (B) Control and shPACT MEFs were transfected with FLAG-NF-κB c-Rel. Cells were fractionated into nuclear and cytoplasmic lysates, and FLAG-NF-κB c-Rel was immunoprecipitated from nuclear lysates. Nuclear extracts and coimmunoprecipitated proteins were analyzed via western blot. (C) Adaptation to increased osmolarity of the extracellular environment involves Rel family transcription factors TonEBP and c-Rel. At high stress intensities, cells fail to adapt and proinflammatory signaling is induced via remodeling of Rel family transcription factor dimer formation. Loss of adaptation to hyperosmotic stress associates with increased TonEBP/p65 dimerization and loss of TonEBP/c-Rel and p65/c-Rel dimerization. PACT-mediated PKR activation marks the loss of adaptation and enhancement of proinflammatory gene expression.

NF-κB was basally higher (12%) than in control cells (1%) prior to low-intensity stress, and increased to a much higher percentage (43%) than in control cells (19%) (*Figure 7A*, middle). Upon loss of PACT, accumulation of c-Rel in the nucleus were greater after low intensity-hyperosmotic treatment (29%) than in control cells (8%) (*Figure 7A*, right). It is important to note that after high intensity hyperosmotic treatments, 86% of total cellular NF-κB c-Rel and over 75% of total NF-κB p65 are found in the cytoplasm. This suggests that NF-κB p65 and c-Rel species may not be the major transcription factors activated to translocate to the nucleus in response to hyperosmotic stress (*Figure 7A*, left). This response differs from other treatments such as LPS or TNFα treatment, in which the vast majority of cellular NF-κB species accumulate in the nucleus upon treatment (*Moreno et al., 2010*; *Li et al., 2017*). The maintenance of NF-κB c-Rel/NF-κB p65 and NF-κB c-Rel/TonEBP complexes as a result of loss of PACT-mediated PKR activation would only be able to influence downstream signaling if these complexes were translocated to the nucleus. To confirm the presence of these complexes in the nucleus, we transfected MEFs with FLAG-tagged NF-κB c-Rel, isolated nuclear fractions after hyperosmotic treatments, and performed co-immunoprecipitation from these nuclear fractions. Despite the majority of NF-κB proteins remaining in the cytoplasm (*Figure 7A*), the interaction between NF-κB c-Rel and TonEBP is still detected in the nucleus in basal conditions and in response to low-intensity hyperosmotic stress (*Figure 7B*, left). The loss of PACT leads to maintenance of these complexes in response to high-intensity hyperosmotic stress (*Figure 7B*, right).

These data suggest that both PKR activation and NF-κB signaling are key components in the determination of cell fate in response to hyperosmotic stress. In low intensity hyperosmotic conditions, TonEBP/NF-κB c-Rel complexes migrate to the nucleus and induce transcription of genes including SNAT2 and GADD34 that contribute to osmoadaptation and cell survival. As intensity increases, PACT will bind to PKR, leading to its phosphorylation and subsequent activation. This correlates with a loss of TonEBP/NF-κB c-Rel and NF-κB p65/NF-κB c-Rel complexes, and instead an accumulation of TonEBP/NF-κB p65 complexes with potential to induce transcription of proinflammatory genes such as IL6 and Tnfrsf9 (*Figure 7C*).

## Discussion

In this study, we have shown that PACT-PKR interaction is necessary for PKR activation and the expression of inflammatory genes and cell death under hyperosmotic conditions. Our findings suggest a mechanism in which PACT-PKR signaling promotes proinflammatory gene expression in a stress intensity-dependent manner, via the modulation of interactions among members of the Rel-homology-domain family of transcription factors. Specifically, we show that in low intensity hyperosmotic stress, NF-κB subunit c-Rel forms complexes with Rel-homology domain transcription factor TonEBP. This interaction preserves adaptive signaling at low-intensity stress via the induced transcription of a key factor in the adaptive response, SNAT2, and its positive regulator, GADD34. The loss of these complexes at high-intensity stress may lead to increased TonEBP interaction with NF-κB p65 and subsequent increased expression of proinflammatory genes such as IL6 and Tnfrsf9. The conservation of this loss of adaptive (*Slc38a2*, *Ppp1r15a*) gene expression indicates conserved expression programs in response to hyperosmotic stress.

The mechanism of PKR activation by PACT is only partially understood. PACT is known to bind to PKR via the N-terminal dsRNA binding domains and the C-terminal kinase domain, though only binding of the kinase domain is necessary for subsequent PKR phosphorylation at Thr-451 and activation (*Peters et al., 2001*). Phosphorylation of PACT at Ser-246 and Ser-287 has been demonstrated to occur constitutively and in response to various cellular stresses such as thapsigargin,

hydrogen peroxide, and sodium arsenite treatment, respectively (*Ito et al., 1999*). We show here that the same sites are likely phosphorylated under hyperosmotic conditions, and are necessary for the activation of PKR. PKR activation via PACT was previously reported in conditions of overexpression of PACT protein, treatment with actinomycin D, growth factor withdrawal (*Marques et al., 2008*), and endoplasmic reticulum (ER) stress (*Singh et al., 2009*). This PACT-mediated activation of PKR in ER stress has been shown to be required for apoptosis after tunicamycin treatment (*Singh et al., 2009*); similarly, PACT phosphorylation at Ser-287 was increased, and association with binding partner TRBP was decreased (*Chukwurah and Patel, 2018*), hinting at an additional mechanism of regulation. To our knowledge this is the first report of PACT-mediated PKR activation in hyperosmotic stress conditions. It is shown here that signaling along the PACT-PKR axis has pro-apoptotic functions during high-intensity hyperosmotic stress. Our findings are in agreement with a human disease state of dystonia, in which an identified PACT mutation was shown to increase association with PKR, correlating with increased apoptosis in patient-derived lymphoblasts (*Vaughn et al., 2015*).

We show here that PKR activation affects the nuclear accumulation of Rel homology domain transcription factors. The exact nature of this mechanism is still unknown. PKR has been shown to directly phosphorylate the inhibitor of κB protein IκBα in vitro and in vivo (*Kumar et al., 1994*), although a loss of PKR does not affect hyperosmotic stress-induced phosphorylation of p65 at Ser-536 (*Farabaugh et al., 2017*). Additionally, the exact mechanism by which PKR signaling affects specific NF-κB dimers in response to hyperosmotic stress remains unclear. It is possible that a subset of NF-κB complexes in the cytoplasm are affected by PACT-PKR signaling via additional interacting or intermediary proteins, and that this subset translocates to the nucleus in response to hyperosmotic stress response. The identification of these complexes and intermediaries will remain for future studies.

PACT when not bound to PKR appears to have some effect on the TonEBP-dependent osmoadaptive gene expression program. Although SNAT2 mRNA is still induced at low stress intensity in PACT-deficient cells, it is induced to a lesser degree (*Figure 1E*), indicating that PACT may be important for osmoadaptation. The mechanisms of the osmoadaptive action of PACT independent of PKR are not known. Because other osmoadaptive mRNAs, such as *Slc6a6* and *Akr1b1* also show reduced induction in PACT-deficient MEFs compared to wild-type MEFs (*Figure 1—figure supplement 2A*), it is suggested that PACT depletion may indirectly inhibit the TonEBP-dependent osmoadaptive program. In agreement with the PACT/PKR signaling inhibiting osmoadaptive gene expression, we also found increased-half lives of *Slc38a2* and *Ppp1r15a* mRNAs during osmoadaptation in PACT-deficient cells (*Figure 1—figure supplement 2B*). Similar data were obtained in PKR deficient cells (data not shown). In contrast to the regulation of the half-life of *Slc38a2* and *Ppp1r15a* mRNAs by PACT availability, the half-life of the *Akr1b1* mRNA was not altered (*Figure 1—figure supplement 2B*). These data suggest that some undetectable levels of PACT-mediated PKR activation plays a role in reducing mRNA stability of the two osmoadaptive genes, potentially via interaction with the PACT-binding partner TRBP (*Chukwurah and Patel, 2018*).

We have shown that the nuclear accumulation of TonEBP is not sufficient to induce transcription of certain osmoadaptive genes in hyperosmotic stress conditions. We show that TonEBP directly forms a complex with NF-κB c-Rel, and that transcription of the osmoadaptive genes *Slc38a2* and *Ppp1r15a* is dependent on NF-κB family transcription factor c-Rel. As both expression of SNAT2 and the NF-κB c-Rel/TonEBP complex are present at low-intensity stress and lost at high-intensity stress, it follows that *Slc38a2* transcription may result from binding of the NF-κB c-Rel/TonEBP complex to the *Slc38a2* promoter. Still outstanding is definitive proof that TonEBP/NF-κB c-Rel complex nuclear accumulation promotes DNA binding and increased transcription, as these aspects can be regulated separately via IκB degradation and re-synthesis (*O'Dea and Hoffmann, 2010*). It is possible that localized DNA characteristics dictate the binding of NF-κB c-Rel/TonEBP complexes to specific gene promoters (*Samee et al., 2019*), or that activation of transcription of these genes is dependent on unique interacting transcription-enhancing proteins rather than DNA binding (*Lee et al., 2016*). In-depth analysis of the binding sites of Rel-homology domain transcription factor heterodimers will need to be performed on genes responsive to NF-κB c-Rel/TonEBP complexes to determine what differences may exist between these and classical tonicity-responsive enhancer sites (TonE sites) known to respond to TonEBP homodimers (*Stroud et al., 2002*).

TonEBP target genes that have been described in the literature include those necessary for osmoadaptation, such as *Slc38a2*, *Akr1b1*, and *Slc6a6*, as well as several that are involved in inflammation, such as *Nos2* (*Lee et al., 2016*), *Ccl2* and *Ptgs2* (*Buxadé et al., 2012*). The combination of TonEBP and NF-κB p65 has been shown to induce transcription of only these inflammatory genes, and not osmoadaptive TonEBP targets (*Johnson et al., 2017*). *Nos2* gene expression was shown to be dependent on the presence of NF-κB p65 (*Farabaugh et al., 2017*), and hyperamplification of its expression correlated with increased levels of nuclear NF-κB p65, as both were increased upon loss of NF-κB c-Rel. This may indicate that controlling the availability of p65 is a critical factor in the amplification of proinflammatory signaling during hyperosmotic stress. Although iNOS was not regulated by TonEBP in response to hyperosmotic stress, other proinflammatory NF-κB targets were induced, including *Il6* and *Tnfrsf9* (*Figure 5D*). It has been shown that while *Nos2* induction is dependent on STAT1 and IRF1 signaling in response to cytokine stimulation, IL6 secretion is independent of this signaling pathway in mouse fibroblasts (*Samardzic et al., 2001*); it is likely that elements in the promoters of these genes exist that lead to differential expression in unique cellular contexts dependent on cell type or stress stimulant.

In this report, we identified the existence of NF-κB c-Rel in complexes with either TonEBP or NF-κB p65 in MEFs. Fibroblasts in particular can be an appropriate model system because not only are they present in all tissues, but they can act as sentinels that can produce inflammatory factors such as Toll-like receptors and cytokines in response to environmental stresses, such as in defense against microbial invasion (*Bautista-Hernández et al., 2017*). This study shows that subsets of genes that share functionality in osmoadaptation and proinflammatory signaling can be targets of specific Rel-homology domain complexes in MEFs, such as TonEBP/NF-κB c-Rel or TonEBP/NF-κB p65, respectively. These complexes were disrupted under high intensity stress conditions in a PACT/PKR-mediated manner. At this high intensity of stress, osmoadaptive genes were not induced, suggesting a duality of TonEBP function. Our data suggest that this duality of function is mediated via formation of complexes with partners, NF-κB p65 or NF-κB c-Rel. It seems plausible that in osmoadaptive conditions, competition between NF-κB c-Rel and NF-κB p65 binding to TonEBP can act as a rheostat between NF-κB c-Rel-mediated osmoadaptive signaling and NF-κB p65-mediated proinflammatory signaling. The existing literature shows anecdotal evidence of NF-κB subunits serving different functions in various cellular contexts: NF-κB c-Rel can have opposing functions to NF-κB p65 in neurons, where NF-κB c-Rel promotes neuronal survival while NF-κB p65 promotes neuronal cell death (*Pizzi et al., 2002*); or c-Rel repressing p65-mediated expression of HIV protein LTR and IL-2Rα (*Doerre et al., 1993*); or c-Rel-driven TRAIL-induced apoptosis rescued by overexpression of p65 (*Chen et al., 2003*). NF-κB subunits can also serve redundant functions; the loss of some individual NF-κB family members can be compensated for by other NF-κB family members, evidenced by deficiency of peripheral lymphoid cells as a result of loss of pro-survival proteins A1 and Bcl-2 in NF-κB p65/NF-κB c-Rel/TNF knockout mice, which is not present in either NF-κB p65/TNF or NF-κB c-Rel knockout mice (*Grossmann et al., 2000*). This is also consistent with genes being targets of specific NF-κB family members only in certain cell types, such as HEK cells having predominantly NF-κB p65/p50 dimers while NF-κB c-Rel is functionally negligible (*O'Dea and Hoffmann, 2010*), or B cells, in which NF-κB c-Rel/p50 is the predominant complex (*Sen, 2006*). The function of these NF-κB c-Rel/TonEBP complexes identified here need to be examined in specific cell types and conditions to be able to directly compare their function with that of other NF-κB complexes.

We have shown here that in high-intensity hyperosmotic stress conditions, osmoadaptation is terminated. However, KEGG pathway analysis of our RNA-seq data in high-intensity stress showed that the most significantly upregulated pathway at high-intensity stress is associated with the ribosome (*Figure 4—figure supplement 1B*), specifically genes that encode ribosomal proteins. It is possible that this increase in transcription is an attempt to recover translation capacity from the sustained reduction in protein synthesis associated with continued stress (*Bevilacqua et al., 2010*). This trend is consistent with the hypothesis that mRNAs transcribed upon high-intensity hyperosmotic stress exposure will be translated upon recovery of cap-dependent protein synthesis upon recovery to iso-osmolarity. Future studies will determine if the cellular response to severe stress involves alternative adaptation mechanisms to return to iso-osmolar conditions.

The novel PACT-PKR signaling pathway expanded upon here will aid our understanding of potential mechanisms involved in hyperactivation of proinflammatory conditions. It is important to note that the influence of PACT-mediated PKR activation on proinflammatory gene expression is only

observed in high-intensity stress. It remains to be determined whether drugs that inhibit PKR activation, including C16 (*Xiao et al., 2016*), or compounds that disrupt PACT/PKR-mediated signaling, such as luteolin (*Dabo et al., 2017*), will be effective in treating diseases with inflammatory components, such as IBD (*Schwartz et al., 2009*), or diseases with a hyperosmotic component, such as dry eye syndrome (*Lemp et al., 2011*). This signaling pathway may also be relevant in certain cancers, such as breast cancer and glioblastoma; inhibition of PKR has been proposed to sensitize Trastuzumab-resistant HER2+ breast cancer to increase efficacy of Trastuzumab in preventing tumorigenesis (*Darini et al., 2019*), and inhibition of NF-κB signaling or iNOS inhibition have been demonstrated to slow growth of glioblastoma stem cells (*Eyler et al., 2011*; *Friedmann-Morvinski et al., 2016*). Proliferation and function of other stem cells, such as hematopoietic stem/progenitor cells and mesenchymal stem cells, are also regulated by PKR activation and inflammatory conditions (*Crop et al., 2010*; *Liu et al., 2013*), and may be relevant model systems to further the exploration of the PACT-PKR signaling pathway.

## Materials and methods

### Cells and reagents

Cell lines used include MEF, HEK293T, RAW, and HLE12 maintained in the lab. Cells were all negative for mycoplasma. Where necessary, knockouts were confirmed by western blot. Mouse embryonic fibroblasts and RAW macrophages were cultured in high glucose Dulbecco's modified Eagle's medium (DMEM) containing fetal bovine serum (FBS, 10%), glutamine (2 mM), penicillin (100 units/ml), and streptomycin (100 μg/ml). Cells were maintained at 37° C and 5% CO$_2$. Early passage HLE12 cells were obtained from Dr. Calvin Cotton, isolated as primary human tracheal epithelial cells from human patients. These cells were cultured in RPMI containing Insulin/transferrin/sodium selenite (Sigma I-1884), transferrin (5 μg/ml), hydrocortisone (10 nM), B-estradiol (10 nM), HEPES (10 mM), glutamine (2 mM), heat-inactivated FBS (4%), penicillin (100 units/ml), and streptomycin (100 μg/ml). Hyperosmotic media was made by adding sucrose to isoosmolar (300 mOsm) media to achieve the desired mOsm, or by adding DSS to achieve the desired weight/volume percentage. Antibodies used include: PACT (from Dr. Ganes Sen); PKR-P(Thr-451) (Santa Cruz sc-101784); PKR (sc-6282); eIF2α-P(Ser-51) (Novus Biologicals NB110-56949); eIF2α (Santa Cruz sc-13327); FLAG (Sigma F1804); TonEBP (from Dr. H.M. Kwon); p65 (Bethyl 301-824A); p65-P(Ser-536) (Cell Signaling 3033); p65-P(Ser-468) (Cell Signaling 3039); c-Rel (Santa Cruz sc-70); histone H3 (Cell signaling 9715); SNAT2 (Medical and Biological Laboratories BMP081); GADD34 (Santa Cruz sc-825); and tubulin (Sigma T9026). Other reagents include DSS (MP Biochemicals, LLC #160110), and PKR inhibitor C16 (Calbiochem #527450). Plasmids used encoded Firefly luciferase-tagged GADD34 (*Crop et al., 2010*), Firefly-luciferase-tagged SNAT2 (from Dr. Simon Kilberg), FLAG-tagged c-Rel (from Dr. Parameswaran Ramakrishnan), FLAG-tagged p65 (*Farabaugh et al., 2017*), and FLAG-tagged wild type and mutant PACT (from Dr. Ganes Sen).

### shRNA

shRNA directed against *Rel* (TRCN0000218674) and *Prkra* (TRCN0000024834) were obtained from Sigma Aldrich. HEK cells were transfected with shRNA constructs and helper plasmids *Pspax2* and *Pmd2g* in a 10:3:2 μg ratio using Xtremegene transfection reagent and protocol (Roche). shControl plasmid consisted of the shRNA-plasmid backbone with puromycin resistance cassette lacking any shRNA target sequence. Media was collected at 12 hr, 36 hr, and 60 hr post-transfection. Collected media was filtered through a 0.45-micron filter syringe to obtain viral particles. MEF cells were then infected with media containing viral particles for 3 days. Cells were selected for infection by puromycin treatment and verified by western blot prior to use.

### Western blotting

Whole cell extracts were prepared by lysing cells in RIPA buffer (100 mM Tris-HCl pH 8.0, 140 mM NaCl, 1% NP-40, 1% sodium deoxycholate, 0.1% SDS, 2 mM EDTA, protease and phosphatase inhibitor tablets (Roche)) on ice. Whole cell extracts were collected after centrifugation at 20,000 g for 15 min at 4° C. Proteins were quantified by Bradford assay and immunoblot analysis was

performed as described previously (*Farabaugh et al., 2017*). Densitometry analysis was performed using Image J.

## Transfection

MEF cells were grown to ~50% confluency on 150 mm plates. 30 µg plasmid was added to an Eppendorf tube with 1 ml OptiMEM media, and 60 µl Xtremegene transfection reagent was added directly to the liquid and mixed by pipetting up and down. After 20 min incubation, transfection media was added to cells dropwise. Media was refreshed after 6 hr, and cells treated with hyperosmotic media after 24 hr.

## Nuclear fractionation

Nuclear extracts were prepared by a two-step lysis process. First, cells were lysed in a hypotonic lysis buffer (10 mM HEPES pH 7.9, 10 mM KCl, 1.5 mM $MgCl_2$, 0.5 mM DTT, 0.25% NP-40, and phosphatase and protease inhibitor tablets) on ice for 10 min. Supernatants containing the cytoplasmic fractions were collected by centrifugation for 4 min at 2500 g at 4° C. Nuclear pellets were washed in lysis buffer lacking NP-40 detergent, and then resuspended in extraction buffer (20 mM HEPES pH 7.9, 0.45 M NaCl, 1 mM EDTA, 0.5 mM DTT, protease and phosphatase inhibitor tablets). Nuclei were lysed by sonication, slowly rotated at 4° C for 30 min, and supernatant was collected after centrifugation at 12,000 rpm for 15 min at 4° C.

## Immunoprecipitations

Protein A/G Dynabeads were prewashed once with NaP (0.1 M, pH = 8.1), and twice with RIPA buffer. 500 µg protein lysate was mixed with beads and 2 µl primary antibody to a volume of 500 µl, and rotated at 4° C for 60 min. Supernatant was removed, and beads were washed once with RIPA buffer. Beads were boiled for 10 min in 1X sample buffer (Thermo Fisher 39001), then immunoprecipitate was collected in a separate tube for analysis via western blot. For immunoprecipitation from nuclear fractions, lysates were diluted in extraction buffer lacking NaCl to balance salt concentration to 140 mM prior to incubation with beads.

## Caspase-3 activity assay

MEFs were treated with hyperosmotic media as described above. Caspase-3 activity was measured as described previously (*Farabaugh et al., 2017*).

## RT-qPCR

Total RNA was prepared using TRIzol reagent (Ambion) according to manufacturer's instructions. RNA was submitted to secondary isolation via LiCl precipitation, as described previously (*Farabaugh et al., 2017*). cDNA was synthesized using Superscript II First-strand synthesis kit (NEB), and analyzed via RT-qPCR using FastStart Universal SYBR Green Master (Roche) as described previously (*Farabaugh et al., 2017*). Data were normalized first to *Gapdh* mRNA, then to unstressed control mRNA levels.

## Proline transport

Proline uptake was measured as described previously (*Krokowski et al., 2015*). Briefly, cells were seeded in 24-well plates at $5 \times 10^4$ cells/well, grown for 24 hr, and subjected to treatment as described above. Cells were washed twice and assays were performed in Earle's balanced salt solution (EBSS) with 100 µM Pro supplemented with [$^3$H]Pro (PerkinElmer; 8 µCi/ml) for 30 s. Amino acids were extracted in ethanol and radioactivity was measured in a liquid scintillation counter. Protein concentration was determined by the Lowry method using BSA as a standard.

## Membrane fractionation

Total membrane fractions were obtained as described previously (*Krokowski et al., 2015*). Cells were washed twice with ice-cold PBS (phosphate-buffered saline) and harvested from culture dishes using a scraper. The cell suspension was centrifuged at 700 × *g* for 10 min and the resultant pellet resuspended in 3 ml of ice cold buffer I (250 mmol/liter sucrose, 20 mmol/liter HEPES, 5 mmol/liter NaN3, 2 mmol/liter EGTA, 100 µmol/liter phenylmethylsulfonyl fluoride; pH 7.4) and homogenized.

The resulting supernatant was centrifuged at 177,000 × *g* for 1 hr at 4°C. The pellet was resuspendend in 50 µl of Buffer I supplemented with protease inhibitor.

## Luciferase assays

Transient transfections were performed using the SNAT2-Firefly luciferase reporter (from Dr. Simon Kilberg) or GADD34-Firefly luciferase reporter (*Krokowski et al., 2015*), and the CMV-Renilla luciferase reporter at 20:1 ratio. Cells were used for subsequent experiments 24 hr after transfection. Luciferase activity was determined using the Dual-Luciferase Reporter System (Promega) as described by the manufacturer.

## RNA-seq and analysis

 MEFs were treated with indicated stress intensities for 3 hr. Cells were pelleted according to company protocol (Novogene), frozen in liquid nitrogen and stored at −80° C. mRNA library preparation and sequencing were carried out by Novogene using poly-T enrichment and buffer fragmentation. 50 Mbp of DNA sequences from 10,443 genes were captured. After DNA quality evaluation, pooled samples were sequenced on Illumina HiSeq 4000 according to the manufacturer's instructions for paired-end 150 bp reads. Reads with adapter contamination, reads containing uncertain nucleotides (>10%), and reads with more than 50% of the read having low-quality nucleotides (Qscore ≥5) were discarded. Clean reads were aligned to the mouse reference genome (UCSC mm10). Differential expression analysis was performed using the DESeq2 R package, and resulting p values were adjusted using the Benjamini and Hochberg's approach for controlling false discovery rate (FDR) (*Anders and Huber, 2010*). Hierarchical clustering analysis was carried out with the log10(FPKM+1) of differentially expressed genes of all comparison groups. Heatmaps were created using the heatmap.2 R package. KEGG pathway analysis was performed on all differentially expressed genes (p<0.05) using DAVID 6.8 (*Huang et al., 2009a*; *Huang et al., 2009b*).

## Statistics

 Statistical analysis was performed using Microsoft Excel. Significant difference between samples was determined using the student's T-test. p values < 0.05 are represented with *, and p values < 0.01 are represented with **.

## Acknowledgements

This body of work was supported by the following grants: NIH R01DK53307, R01DK060596, and R01DK113196 to MH; NIH/NIAID grants R01AI116730 and R21AI144264 to PR; CDDRCC pilot grant DK097948 to MH; and National Science Centre (Poland) 2018/30/E/NZ1/00605 to DK.

## Additional information

### Funding

| Funder | Grant reference number | Author |
| --- | --- | --- |
| National Institutes of Health | R01DK53307 | Maria Hatzoglou |
| National Institute of Allergy and Infectious Diseases | R01AI116730 | Parameswaran Ramakrishnan |
| National Science Centre | 2018/30/E/NZ1/00605 | Dawid Krokowski |
| Cleveland Digestive Disease Research Core Center | DK097948 | Maria Hatzoglou |
| National Institutes of Health | R01DK060596 | Maria Hatzoglou |
| National Institutes of Health | R01DK113196 | Maria Hatzoglou |
| National Institute of Allergy and Infectious Diseases | R21AI144264 | Parameswaran Ramakrishnan |

The funders had no role in study design, data collection and interpretation, or the decision to submit the work for publication.

## Author contributions
Kenneth T Farabaugh, Raul Jobava, Greeshma Ray, Parameswaran Ramakrishnan, Maria Hatzoglou, Conceptualization, Resources, Data curation, Formal analysis, Supervision, Funding acquisition, Validation, Investigation, Visualization, Methodology, Project administration; Dawid Krokowski, Data curation, Formal analysis, Funding acquisition, Validation, Investigation, Methodology; Bo-Jhih Guan, Validation, Investigation, Methodology; Zhaofeng Gao, Conceptualization, Data curation, Software, Formal analysis, Validation, Investigation, Visualization, Methodology, Project administration; Xing-Huang Gao, Data curation, Software, Formal analysis, Investigation, Methodology; Jing Wu, Massimiliano G Bianchi, Ovidio Bussolati, Michelle Longworth, Data curation, Formal analysis, Investigation, Methodology; Tristan J de Jesus, Conceptualization, Resources, Data curation, Supervision, Funding acquisition, Validation, Investigation, Visualization, Methodology, Project administration; Evelyn Chukwurah, Data curation; Michael Kilberg, Calvin Cotton, Resources; David A Buchner, Data curation, Formal analysis; Ganes C Sen, Data curation, Investigation, Methodology; Christine McDonald, Formal analysis

## Author ORCIDs
Kenneth T Farabaugh ⬚ https://orcid.org/0000-0002-9591-0466
Xing-Huang Gao ⬚ http://orcid.org/0000-0002-0720-3690
Ovidio Bussolati ⬚ http://orcid.org/0000-0002-4301-2939
David A Buchner ⬚ http://orcid.org/0000-0003-3920-4871
Parameswaran Ramakrishnan ⬚ https://orcid.org/0000-0002-1314-827X
Maria Hatzoglou ⬚ https://orcid.org/0000-0003-2037-1231

## Ethics
Animal experimentation: This study was performed in strict accordance with the Guide for the Care and Use of Laboratory Animals of the National Institutes of Health. All of the animals were handled according to approved institutional animal care and use committee (IACUC) protocols (#400061) of Case Western Reserve University.

## Decision letter and Author response
Decision letter https://doi.org/10.7554/eLife.52241.sa1
Author response https://doi.org/10.7554/eLife.52241.sa2

# Additional files

## Supplementary files
• Transparent reporting form

## Data availability
Sequencing data have been deposited in GEO under accession code GSE138692.

The following dataset was generated:

| Author(s) | Year | Dataset title | Dataset URL | Database and Identifier |
|---|---|---|---|---|
| Farabaugh KT, Hatzoglou M | 2019 | RNA-sequencing in Mouse Embryonic Fibroblasts exposed to different Intensities of Hyperosmotic Stress | https://www.ncbi.nlm.nih.gov/geo/query/acc.cgi?acc=GSE138692 | NCBI Gene Expression Omnibus, GSE138692 |

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
