## [Decision Letter]

**Acceptance summary:**

Hyperosmotic stress contributes to pathological outcomes in complex diseases such as diabetes and inflammatory bowel disease. This work provides compelling evidence for a molecular switch that flips cellular programming from a protective adaptive response to osmotic stress towards an inflammatory response. These findings advance our understanding of how cells translate physicochemical stress into inflammation and identify potential pathways for therapeutic intervention.

**Decision letter after peer review:**

Thank you for submitting your article "PACT-PKR activation acts as a hyperosmotic stress intensity sensor weakening osmoadaptation and enhancing inflammation" for consideration by *eLife*. Your article has been reviewed by two peer reviewers, and the evaluation has been overseen by a Reviewing Editor and Tadatsugu Taniguchi as the Senior Editor. The reviewers have opted to remain anonymous.

The reviewers have discussed the reviews with one another and the Reviewing Editor has drafted this decision to help you prepare a revised submission.

Summary:

This study identifies PACT as a key determinant driving the inflammatory response to hyperosmotic stress. The mechanisms that link hypertonicity to inflammation in disease are poorly defined and this work has the potential to advance the fields of cellular stress and inflammation. While previous studies established that PKR was activated under hyperosmotic stress conditions, stimulating inflammation through NF-κB, these new data provide evidence that the RNA-binding protein, PACT, interacting with PKR, enhances transcriptional programming favoring pro-inflammatory signaling at the expense of osmoadaptation. The pro-inflammatory transcriptional response to hyperosmotic stress resulted from preferential formation of TonEBP/NF-κB p65 heterodimers promoted by the PACT-PKR complex, rather than TonEBP/NF-κB c-Rel. The studies also highlight the function of c-Rel in the adaptive response to hyperosmotic stress at low stress levels, and that higher stress levels cause a change in the subunit compositions of the transcription factors belonging to the Rel family. Collectively, these data support a model that connects hyper-osmotic stress to pro-inflammatory responses through the PACT-PKR complex, identifying this complex as a potential target in diseases such as inflammatory bowel disease or dry eye syndrome that involve osmotic imbalance.

Essential revisions:

1) The rigor of the study would be improved by ChIP analysis with c-Rel, TonEBP, and p65 RelA antibodies on endogenous promoters under conditions of low and high stress to demonstrate more directly the association of the relevant transcriptional regulators with target genes in high and low osmotic stress conditions. This is important because shifts in dimeric compositions of the Rel family transcription factors that occur from low stress to high stress conditions are demonstrated only under artificial overexpression conditions.

2) Substantial reorganization of the manuscript to include the following:

a) Addressing concerns about the claims based on data in Figure 1E. The SNAT2 mRNA levels are significantly lower at low intensity stress in shPACT cells. The authors do not present a convincing argument for this observation. Is SNAT2 mRNA regulated at mRNA stability level by PACT-PKR axis?

The key claim of the authors maintains that PACT/PKR signaling impairs osmoadaptation. In Figure 1E they show, however, that the levels of the osmoadaptive response gene SNAT2 are independent of PACT. This questions their key claim. In line with this, the authors state that Nos2 belongs to the group of inflammatory genes induced by high osmotic stress. However, in Figure 5 it is surprising that Nos2 does not any more belong to this group. This warrants an explanation.

b) Overall, it was felt that the assessment of c-Rel in the DSS in vivo model does not substantiate the in vitro findings on hyperosmotic stress but is a separate story. Therefore, consider removing the c-Rel in vivo data as it does not contribute strongly to the conclusions of the manuscript in its current form.

c) Revise the text with attention to the specific suggestions, but also with detailed attention to proofreading and a more concise structure.

[Editors' note: further revisions were suggested prior to acceptance, as described below.]

Thank you for resubmitting your work entitled "PACT-PKR activation acts as a hyperosmotic stress intensity sensor weakening osmoadaptation and enhancing inflammation" for further consideration by *eLife*. Your revised article has been evaluated by Tadatsugu Taniguchi (Senior Editor) and a Reviewing Editor.

The manuscript has been improved but there are some remaining issues that need to be addressed before acceptance, as outlined below:

Overall, the revised manuscript provides compelling evidence for a PACT-PKR dependent mechanism that controls switching between an osmoadaptive and an inflammatory mechanism. We appreciate that you have tried to address a major concern of the reviewers, namely direct evidence for c-Rel binding, and understand that technical challenges will prevent including such data in the current manuscript. Accordingly, please clarify specific wording in the manuscript that might imply that c-Rel acts directly as a transcription factor, since that will likely be an assumption readers will make.

For example, "We show that TonEBP directly forms a complex with NF-κB c-Rel, and that osmoadaptive genes *Slc38a2* and *Ppp1r15a* being targets of NF-κB family transcription factor c-Rel." This sentence should more accurately reflect that *Slc38a2* and Ppp1r5a transcription is dependent on c-Rel, since it is not yet clear if they are direct targets. In general, the text was carefully worded to avoid implying direct binding by c-Rel, but there may be other instances. In general, this is an exciting finding and with these minor changes will be an important addition to the field.

---

## [Author Response]

Essential revisions:1) The rigor of the study would be improved by ChIP analysis with c-Rel, TonEBP, and p65 RelA antibodies on endogenous promoters under conditions of low and high stress to demonstrate more directly the association of the relevant transcriptional regulators with target genes in high and low osmotic stress conditions. This is important because shifts in dimeric compositions of the Rel family transcription factors that occur from low stress to high stress conditions are demonstrated only under artificial overexpression conditions.

We have attempted in the past ChIP analysis with both c-Rel and p65 antibodies, unfortunately with limited success. In further attempts to achieve this data, we enlisted the aid of a professional research company, Active Motif, to carry out the experiment as well. Active Motif was also unable to provide this data, which may indicate that the particular combination of our experimental conditions and the available reagents are not suitable to attain these results.

For the reviewers only, we provide in Author response image 1 and Author response image 2 sample of the data we received from Active Motif, demonstrating that ChIP-seq reads with either c-Rel or p65 antibodies were not enriched with signal, in comparison to a histone marker (H3K4me3) positive control. Despite relaxing the cutoff for peak identification, few peaks were found for c-Rel and p65 above background noise. The genes identified were not enriched for NF-κB targets and the common sequences bore no resemblance to the κB consensus binding sequence. Our future work will include elaborate optimization of the ChIP conditions to study stress-induced NF-κB activation, both in house as well as through research company collaborations.

**Author response image 1. respfig1:** Number of ChIP-seq peaks for regions actively binding c-Rel were low, indicating failure of experiment.

**Author response image 2. respfig2:** Number of ChIP-seq peaks for regions actively binding p65 were low compared to positive control H3K4me3, indicating failure of experiment.

2) Substantial reorganization of the manuscript to include the following:a) Addressing concerns about the claims based on data in Figure 1E. The SNAT2 mRNA levels are significantly lower at low intensity stress in shPACT cells. The authors do not present a convincing argument for this observation. Is SNAT2 mRNA regulated at mRNA stability level by PACT-PKR axis?The key claim of the authors maintains that PACT/PKR signaling impairs osmoadaptation. In Figure 1E they show, however, that the levels of the osmoadaptive response gene SNAT2 are independent of PACT. This questions their key claim. In line with this, the authors state that Nos2 belongs to the group of inflammatory genes induced by high osmotic stress. However, in Figure 5 it is surprising that Nos2 does not any more belong to this group. This warrants an explanation.b) Overall, it was felt that the assessment of c-Rel in the DSS in vivo model does not substantiate the in vitro findings on hyperosmotic stress but is a separate story. Therefore, consider removing the c-Rel in vivo data as it does not contribute strongly to the conclusions of the manuscript in its current form.c) Revise the text with attention to the specific suggestions, but also with detailed attention to proofreading and a more concise structure.

a) The reviewer is correct that SNAT2 mRNA levels display less of an increase in shPACT MEFs at the lower stress intensity (500 mOsm). The reviewer argues that this is in contrast to the authors’ claim that PACT-PKR signaling only affects osmoadaptation at high stress intensity (600 mOsm) when PACT-PKR interaction is observed. To address the possibility that PACT affects the level of SNAT2 mRNA at the level of mRNA stability, we performed an RNA stability assay using Actinomycin D after 3 h treatment with 500 mOsm stress media. The results (Figure 1—figure supplement 2B) actually showed that both SNAT2 and GADD34 mRNA stability was increased in PACT KO MEFs while AR mRNA stability was not, suggesting that PACT may contribute to increased degradation of the SNAT2 and GADD34 mRNAs during osmoadaptation, via direct or indirect mechanisms. In addition, we obtained similar data on the half-lives of these mRNAs from PKR KO cells treated with 500 mOsm. Both SNAT2 and GADD34 mRNA stability was increased in PKR KO MEFs (data not shown). This data is in agreement with the effects of PACT/PKR being inhibitory in the osmoadaptation program, which is the conclusion of this manuscript. It is possible that small amounts of PACT-mediated PKR activation that are not observed by co-IP contribute to this degradation. It is known that PACT can play a role in mRNA stability via interaction with RNA-binding protein TRBP. It will remain for future studies to determine whether this PACT/TRBP interaction plays a role in stability of osmoadaptive mRNAs in hyperosmotic stress conditions. The mechanism via which the PACT/PKR signaling controls mRNA stability during osmoadaptation will be the focus of our future studies.

As the stability of SNAT2 mRNA was not lessened by knockout of PACT, we hypothesized that PACT may instead affect transcription of the general TonEBP osmoadaptive program. RT-qPCR showed that levels of AR and TauT, also TonEBP targets, were similarly reduced at the low stress intensity in PACT KO MEFs (Figure 1—figure supplement 2A). We conclude that PACT likely has some effects on the adaptive TonEBP-dependent transcriptional program via unknown mechanisms.

We have shown that iNOS mRNA levels do rise with intensity of hyperosmotic stress by RT-qPCR. The RNA-Seq data also show a similar trend in iNOS mRNA induction; however, several factors prohibit this data from being included in the manuscript. The raw number of reads for iNOS mRNA was very low, such that the fpkm only increased from 0.003 to 0.017. In addition, the variability among the replicates results in very large error bars, and very high p values.

**Author response image 3. respfig3:** Induction of Nos2 mRNA was too variable and had too low FPKM to be statistically significant.

b) We have removed the in vivodata in c-Rel-deficient mice.

c) The entire text has been proofread.

[Editors' note: further revisions were suggested prior to acceptance, as described below.]

Overall, the revised manuscript provides compelling evidence for a PACT-PKR dependent mechanism that controls switching between an osmoadaptive and an inflammatory mechanism. We appreciate that you have tried to address a major concern of the reviewers, namely direct evidence for c-Rel binding, and understand that technical challenges will prevent including such data in the current manuscript. Accordingly, please clarify specific wording in the manuscript that might imply that c-Rel acts directly as a transcription factor, since that will likely be an assumption readers will make.For example, "We show that TonEBP directly forms a complex with NF-κB c-Rel, and that osmoadaptive genes Slc38a2 and Ppp1r15a being targets of NF-κB family transcription factor c-Rel." This sentence should more accurately reflect that Slc38a2 and Ppp1r5a transcription is dependent on c-Rel, since it is not yet clear if they are direct targets. In general, the text was carefully worded to avoid implying direct binding by c-Rel, but there may be other instances. In general, this is an exciting finding and with these minor changes will be an important addition to the field.

The newly revised manuscript contains the following changes:

1) The sentence has been changed to read as follows: “These data suggest that NF-κB c-Rel is a novel factor in the transition from adaptation to proinflammatory gene expression programs with increasing stress intensity.”

2) The sentence has been changed to read as follows: “To our knowledge, the dependence of osmoadaptation on NF-κB c-Rel is a new finding in the literature.”

3) The sentence has been changed to read as follows: “These data suggest that both PKR activation and NF-κB signaling are key components in the determination of cell fate in response to hyperosmotic stress.”

4) The sentence has been changed to read as follows: “This correlates with a loss of TonEBP/NF-κB c-Rel and NF-κB p65/NF-κB c-Rel complexes, and instead an accumulation of TonEBP/NF-κB p65 complexes with potential to induce transcription of proinflammatory genes such as IL6 and Tnfrsf9 (Figure 7C).”

5) The sentence has been changed to read as follows: “Our findings suggest a mechanism in which PACT-PKR signaling promotes proinflammatory gene expression in a stress intensity-dependent manner, via the modulation of interactions among members of the Rel-homology-domain family of transcription factors.”

6) The sentence has been changed to read as follows: “We show that TonEBP directly forms a complex with NF-κB c-Rel, and that transcription of the osmoadaptive genes *Slc38a2* and *Ppp1r15a* is dependent on NF-κB family transcription factor c-Rel.”

7) The sentence has been changed to read as follows: “It seems plausible that in osmoadaptive conditions, competition between NF-κB c-Rel and NF-κB p65 binding to TonEBP can act as a rheostat between NF-κB c-Rel-mediated osmoadaptive signaling and NF-κB p65-mediated proinflammatory signaling.”